

# Marine heatwaves deeply alter marine food web structure and function

Vianney Guibourd de Luzinais[1,2*], William W. L. Cheung[2], Didier Gascuel[1]

[1]UMR Dynamics and sustainability of ecosystems: from source to sea (DECOD), Institut Agro, Ifremer, INRAE, Rennes, France
[2]Institute for the Oceans and Fisheries, the University of British Columbia, Vancouver, British Columbia, Canada

*Correspondence to*: Vianney Guibourd de Luzinais (vianney.guibourddeluzinais@institut-agro.fr)

## 1. Abstract

Marine heatwaves (MHWs) are becoming longer, more frequent and more intense in recent decades. MHWs have caused large-scale ecological impacts, such as coral bleaching, mass mortality of seagrass, fishes and invertebrates, and shifts in abundance and distribution of marine species. However, the implications of these MHW-induced impacts on marine species for the structure and functioning of marine food webs are not clearly understood. In this study, we use the EcoTroph-Dyn ecosystem modelling approach to examine the impacts of MHWs occurring during the year's warmest month on the trophodynamics of marine ecosystems. EcoTroph-Dyn represents marine ecosystem dynamics at a spatial resolution of 1° longitude by 1° latitude and a temporal resolution of 14 days. We applied the model to simulate changes in trophodynamic processes, energy transfer and ecosystem biomass using daily temperature and monthly net primary production (NPP) that were derived from satellite observation from 1998 to 2021. We compared and contrasted the simulated changes in biomass by trophic levels with results generated from temperature and NPP time series that had been filtered to remove MHWs. Our results show a significant decline in biomass between 1998 and 2021 specifically caused by MHWs. For example, in the Northeast Pacific Ocean, our model simulated a specific MHW decline in biomass of 8.7% $\pm$ 1.0 (standard error) in the region from 2013 to 2016. Overall, MHW-induced biomass declines are more pronounced in the northern hemisphere and Pacific Ocean. Moreover, the MHW-induced declines in high trophic level biomass were larger than lower trophic levels and lasted longer post-MHW. Finally, this study highlights the need to integrate MHWs into modelling the effects of climate change on marine ecosystems. It shows that the EcoTroph approach, and especially its new dynamic version, provides a framework to understand more comprehensively the implications of climate change for marine ecosystem structure and functioning.



## 2.  Introduction

Over the last century, marine heatwaves (MHWs), defined as discrete periods of anomalously warm water, have increased in frequency, duration, and intensity (Frölicher and Laufkötter, 2018; Oliver et al., 2018). Between the 1920s and the 2010s, their duration and frequency increased by 17% and 34%, respectively, resulting in more than doubling the number of MHWs days at a global scale(Oliver et al., 2018). Since the 2000s, MHWs that are spatially and temporally extensive have been recorded in the world oceans, such as the North Pacific MHW in 2013-2015 (Bond et al., 2015; Cavole et al., 2016), the Mediterranean Sea 2003 and 2022 MHW (Garrabou et al., 2009, 2022), the Western Australia MHW in 2011 (Pearce et al., 2011), and the Tasman Sea MHW in 2015-2016 (Oliver et al., 2017). Documented ecological consequences of these MHW events include coral bleaching (Garrabou et al., 2009, 2022; Pearce et al., 2011) and ecosystem "tropicalization" (Wernberg et al., 2016). Also, MHWs caused a decrease in phytoplankton biomass and diversity that has led to significant changes in zooplankton and other marine invertebrate diversity and biomass (Cavole et al., 2016), and mass mortality of fish and invertebrates (Cannell et al., 2019; Cavole et al., 2016; Collins et al., 2019). However, these ecological impacts of MHWs are not ubiquitous and varied largely between MHW events (Fredston et al., 2023; Oliver et al., 2021; Pershing et al., 2018; Smale et al., 2019; Smith et al., 2023).

The ecological impacts of MHWs have predominantly been studied using laboratory experimentations, analyses of ecological time series, and simulation modelling (Joyce et al., 2024). For example, Carneiro et al., (2020) assessed the evolution of physiological and biochemical parameters and survival rates of the clam *Anomalocardia flexuosa* in response to simulated MHWs. Fredston et al., (2023) studied the effects of MHWs on marine species biomass by analysing scientific trawl survey data (FishGlob data) and historical temperature records. Cheung et al., (2021) and Cheung & Frölicher, (2020) employed marine ecosystem models to investigate MHW implications for biomass and potential catches of exploited marine species and their implication for fisheries. They found that MHWs may cause biomass decreases and shifts in the biogeography of fish stocks that are faster and bigger in magnitude than the effects of decadal-scale mean changes. They projected a doubling of impact levels by 2050 amongst the most important fisheries species over previous assessments that focus only on long-term climate change.

Trophic dynamics of marine ecosystems are affected by ocean temperature (Eddy et al., 2021; du Pontavice et al., 2020).  In particular, ocean warming affects biomass transfer efficiency and the speed of energy transfer through the food web, which results in declines in marine animal biomass (du Pontavice et al., 2021; Guibourd de Luzinais et al., 2023). We expect that temperature changes during



MHWs will also impact trophodynamics and marine animal biomass. However, the effects of MHWs on ecosystem structure and function globally have not yet been clearly understood.

This study aimed to disentangle the additional or synergistic consequences of MHWs occurring during the year's warmest month with the effects of the slow-onset climate changes in marine ecosystems. I used the EcoTroph-Dyn ecosystem model that was developed and applied to study the

responses of marine ecosystems to MHW (see (Guibourd de Luzinais et al., 2024). We explored the effects of different scenarios of MHW-induced community mortality that were dependent on climatic conditions and species' resistance capacities. We undertook a global-scale hindcast analysis over the 1998 to 2021 period and analysed the added impacts of MHWs on marine ecosystem biomass and trophodynamics under climate change. Subsequently, we delved into distinct geographical

characteristics and identified marine ecosystems that are particularly sensitive to MHW-induced biological impacts.  Last but not least, as a case study, we examined a recognised MHW ('the Blob', which occurred on the Northeast Pacific Ocean from 2013 to 2016) and highlighted how MHWs can trigger long-term changes in the ecosystem proceeding to their occurrences.

## 3.    Material and method

### 3.1  The EcoTroph dynamic model

We used EcoTroph-Dyn, a dynamic version derived from the steady-state EcoTroph trophodynamic modelling approach first proposed by Gascuel, (2005) and further developed by Gascuel & Pauly, (2009). In EcoTroph, biomass is produced from primary producers (trophic level TL = 1) and consumed by heterotrophic organisms (TL>1). Thus, the food web functioning is represented as

a continuous flow of biomass surging up the food webs, from primary producers (low TLs) to top predators (high TLs). The resulting ecosystem structure is a continuous distribution of biomass along a gradient of TLs i.e., biomass trophic spectra (Gascuel et al., 2005, Figure 1). Practically, each biomass trophic spectra with TL above 1 is split into small trophic classes bounded by pre-defined lower and upper trophic levels (with conventionally TL width = 0.1 in the steady-state version of EcoTroph). The

biomass spectrum in EcoTroph generally refers to whole consumers biomass, including organisms living in pelagic, mid-water, and benthic habitats. The time needed for the biomass to flow from one to the next trophic class varies along the food chain, with biomass transfers generally faster in lower TLs than in the higher ones.

EcoTroph has been applied to study the long-term effects of fishing, e.g., du Pontavice et al.,

2023; Gasche et al., 2012; Gasche & Gascuel, 2013; Halouani et al., 2015; Tremblay-Boyer et al., 2011 and climate change (du Pontavice et al., 2021) on ecosystems biomass and production. EcoTroph-Dyn, the dynamic version of EcoTroph, simulates changes in biomass flows in the trophic biomass spectra



at a 2-week (14 days) time step. This time step was used as it represents the average duration of most naturally occurring or experimentally simulated MHWs (Smale et al., 2015). EcoTroph-Dyn includes

algorithms that model changes in flow kinetics, boundaries of trophic classes and the resulting biomass states and flow between trophic classes, and an algorithm that models the biomass loss associated with MHWs occurrence. EcoTroph-Dyn algorithms' details are described in Guibourd de Luzinais et al., 2024. Here, we provided a summary of these algorithms.

The quantity of biomass flowing from a trophic class to the next, due to predation or ontogeny,

is not conservative and can be calculated at each time step as:

$$\Phi_{\tau+\Delta\tau,t+1} = \Phi_{\tau,t} \cdot e^{-(\mu_t+\eta_{\tau,t})\cdot\Delta\tau_t}, \quad (1)$$

where $\mu_\tau$ (expressed in $TL^{-1}$) represents the mean natural losses (due to non-predation mortalities, excretion and respiration), which defines the proportion of biomass that does not stay in or move up in the food web (Gascuel et al., 2008; du Pontavice et al., 2020). The $\eta_{\tau,t}$ parameter (expressed in $TL^{-1}$)

represents the additional loss rate specifically due to mortalities induced by MHW events (section 3.2.2).

The flow kinetic $K_\tau$ (expressed in TL.year$^{-1}$), representing the speed of biomass flows (trophic transfers) from low to high TLs, is inversely proportional to biomass residence time (the time each biomass unit stays at a given TL). $K_\tau$ is expressed as a function of trophic level, using the empirical

equation described in Gascuel et al., (2008):

$$K_\tau = 20.19 \cdot \tau_j^{-3.258} \cdot e^{0.041 \cdot SST_y}, \quad (2)$$

where $SST_y$ corresponds to the annual moving average sea surface temperature in year y and $\tau_j$ corresponds to the trophic level of each TL classes j.

EcoTroph-Dyn assumes that MHWs can increase the mortality of marine organisms and thus reduce

their life expectancy. Therefore, according to equations used in the steady-state version of EcoTroph, the flow kinetics during MHWs is increased as follows:

$$K_\tau = 20.19 \cdot \tau_j^{-3.258} \cdot e^{0.041 \cdot SST_y} \cdot (1+\eta_\tau), \quad (3)$$

where $\eta_\tau$ is the MHW-associated additional loss rate.

Biomass spectra in EcoTroph-Dyn are split into trophic classes with variable widths of trophic levels. The width of trophic classes [$\tau_j$; $\tau_{i+1}$[ was determined based on the estimated mean flow kinetics so that biomass could transfer up a trophic class in each time step (Δt = 1/24 year). Thus:

$$\Delta\tau = \tau_{j+1} - \tau_j = K\tau \cdot \Delta t = 20.19 \cdot \tau_j^{-3.258} \cdot e^{0.041 \cdot SST_y} \cdot (1+\eta_\tau) \cdot \Delta t, \quad (4)$$

where $\eta_\tau$, the MHW-associated additional loss rate, is only considered during MHWs.

According to fluid dynamic equations, the EcoTroph-Dyn state variable $B_{\tau,t}$, representing the biomass present at time step t within the TL class [τ,τ+Δτ[. It is expressed as:



$$B_{\tau,t} = \frac{1}{K_{\tau,t}} \cdot \Phi_{\tau,t} \cdot \Delta\tau \ , \quad (5)$$

where $\Phi_{\tau,t}$ is the mean quantity of biomass flowing within the trophic class [τ, τ + Δτ[ at time step t,

and $K_{\tau,t}$ is the mean flow kinetic within the trophic class [τ, τ + Δτ[.

Finally, the production $P_{\tau,t}$ of the trophic class [τ, τ + Δτ[ at time step t is:

$$P_{\tau,t} = \int_{s=\tau}^{s=\tau+\Delta\tau} \Phi(s,t) \cdot ds = \Phi_{\tau,t} \cdot \Delta\tau \ , \quad (6).$$

Hence, according to equations (5) and (6), EcoTroph highlights that biomass stems from the

ratio of the production to the flow kinetic. Production is expressed in t·TL·year$^{-1}$, that is, biomass in

weight moving up the food web by 1 TL on average during one year.

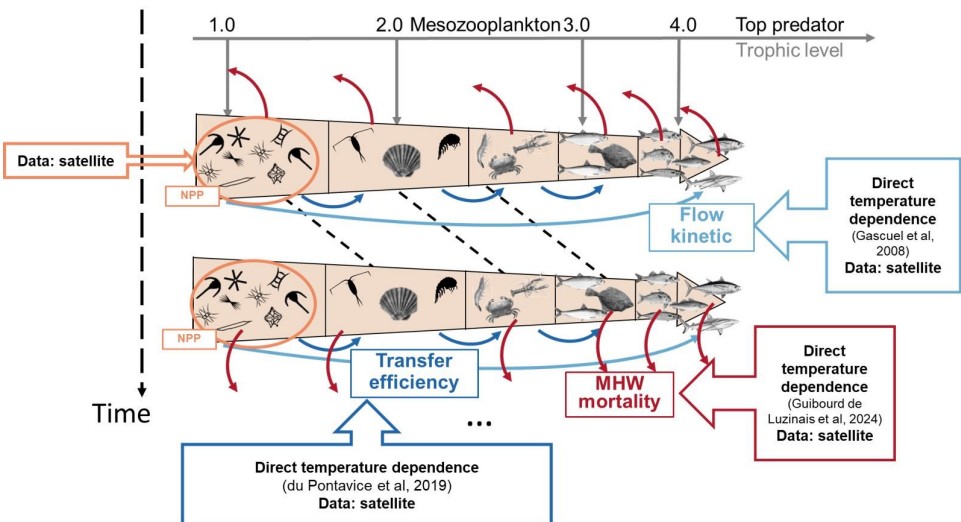

**Figure 1: Schematic diagram illustrating the ecological framework represented by EcoTroph-Dyn (adapted from du Pontavice et al., 2021).** The trophic functioning of marine food webs is represented by a biomass flow, starting with biomass entering the system at trophic level 1 through net primary production (NPP). Biomass then moves through each trophic level according to trophic transfer efficiency. The flow kinetic, temperature and MHW dependant, is crucial in defining trophic class boundaries and estimating biomass for each trophic level over time. When MHWs occur, there is a loss of biomass flow at each trophic level. The empirical models used for each parameter (MHW losses (η), transfer Efficiency (TE), and flow kinetic (K) and where satellite data are incorporated into the model to account for marine heatwaves are noted in the figure.

### 3.2 MHW loss rate algorithm computation

We compute the loss rate in the biomass flow associated with MHWs based on the percentage

of species undergoing thermal stress, i.e., species exposed to temperatures exceeding their thresholds.

A detailed description of the MHW loss rate algorithm computation is described in S1. Here, we

provided the main steps for the incorporation of the effects of marine heatwaves in EcoTroph-Dyn:

1. Detection and characterisation of MHWs with climatology period (1982-2011)

2. Estimation of species distribution and associated thermal niche

3. Matching MHWs' historical distributions and characteristics with species distribution





4. Calculation of the percentage of species undergoing thermal stress in each ocean spatial cell

5. Estimation based on this percentage of an additional loss rate associated with MHWs

Finally, through loss rate (ηi) mathematical expression, we assumed that species were continuously challenged by MHW increased intensity expressed as:

$$\eta_i = e^{-e^{b\_tl_i \cdot \alpha \cdot \left(MHW_{cat,i} - \frac{lt50\_tl_i}{\alpha}\right)}} \cdot \beta, \, (7)$$

where $b\_tl_i$ and $lt_{50}\_tl_i$ correspond to the slope of the function and the index of marine heatwave intensity ($MHW_{cat,i}$) at which 50% of the species are undergoing thermal stress in $cell_i$, $\beta$ corresponds to the MHW duration, with $\beta$ ranges from $\beta=0$; no MHW, $\beta=1$; MHW lasting 15 days of the fortnight (see section 3.3.1). $MHW_{cat,i}$ corresponds to MHW intensity index (defined in S1) and $\alpha$ is a coefficient assumed to represent the species' resistance capacity to MHW conditions, thus reducing the mortality rate due to species' exposure to thermal stress.

An $\alpha = 0.2$ has been considered the central hypothesis used in our simulations. However, we also explored the sensitivity of the results to community resistance capacity to MHW by testing four alternative values of $\alpha$. The $\alpha$ values that we tested were full resistance ($\alpha = 0$, i.e., we fixed $\eta_i = 0$; no mortality due to thermal stress), partial resistance ($\alpha = 0.2, 0.5$; 20%, and 50% of the species die because of thermal stress, respectively) and no resistance ($\alpha = 1$; all species die when they are under thermal stress). We also related the loss rate to the MHW duration over the fortnight by assuming that the duration increased the mortality rate.

### 3.3 EcoTroph-Dyn simulations

The EcoTroph-Dyn model was applied to simulate consumer biomass and production (between TLs 2 and 5.5) from 1998 to 2021 for each 1° × 1° spatial cell of the global ocean (Appendix S2). Model outputs were summarised spatially by biomes: tropical, temperate, or upwelling biomes, based on Reygondeau et al., (2013). We excluded polar biomes because net primary production (NPP) data, one of the forcing variables of the EcoTroph-Dyn model, were not available in high-latitude regions during the winter period.

#### 3.3.1 Environmental forcing data

We simulated and compared consumer biomass and production under scenarios with and without MHW. The simulations were driven by daily SST observations from the National Oceanic and Atmospheric Administration - Advanced Very High-Resolution Radiometer (NOAA _AVHRR) data (https://www.ncei.noaa.gov/access/metadata/landing-page/bin/iso?id=gov.noaa.ncdc:C00680) and NPP data predicted from the EPPLEY-VGPM algorithm and satellite remote sensing data (http://orca.science.oregonstate.edu/npp_products.php).



From the daily SST time series in each ocean cell, we identified every MHWs day. We defined
an MHW as when the daily SSTs exceed an extreme temperature threshold value for at least five
consecutive days (Hobday et al., 2016). The extreme temperature threshold value was calculated for
each 1° latitude x 1° longitude spatial cell as the 90th percentile of daily SST from the 30-year historical
time series from January 1982 to December 2011. We did not calculate threshold values by season;
thus, MHW events were identified by a single threshold across the year. As a result, we detected MHWs
mostly occurring during the year's warmest months. We used the R package heatwaveR described at
https://robwschlegel.github.io/heatwaveR/ (Schlegel and Smit, 2018) to detect MHWs in each spatial
cell from January 1998 to December 2021.

For the scenarios 'with MHWs', we used the SST time series from NOAA _AVHRR data.

To simulate scenarios without MHW in the SST time series, we decomposed the daily SST time
series ($Y_t$) of each ocean spatial cell using a Census X-11 procedure (Pezzulli et al., 2005; Shiskin, 1967;
Vantrepotte and Mélin, 2011). With this method, the time series can be decomposed as:

$$Y_t = T_t + S_t + H_t \ ,$$

where $Y_t$ represents the daily SST at day t, $T_t$ represents the long-term mean annual changes in
temperature, $S_t$ represents the seasonal component (repetitive pattern over time), and $H_t$ represents
the additional temperature variability that is not attributed to the annual trend or seasonality.

The $T_t$ underlying long-term direction is obtained from the 365-day running average of the initial series
$Y_t$. The seasonal component ($S_t$), is then computed by applying a centered moving average to the
trend-adjusted series ($Y_t – T_t$). The estimation of ($S_t$) on the trend-adjusted series avoids any confusion
with the inter-annual (trend) signal. After the revised estimation of these two components (see the
method in Pezzulli et al., 2005; Vantrepotte & Mélin, 2011), The additional temperature variability
component was computed as $H_t = Y_t - S_t - T_t$.

We applied the following procedure to create a daily SST time series for the scenarios without
MHW. When the daily $Y_t$ value was identified as MHW and $Y_t$ was above the value ($T_t + S_t$), we
replaced the daily SST value ($Y_t$) with the expected temperature without the additional variability ($H_t$)
component i.e., $T_t + S_t$. For MHW days with $Y_t$ below ($T_t + S_t$) or not an MHW day, we keep the daily
SST value $Y_t$ (see S3, where the process of creating this time series is illustrated). The time series
created using this algorithm is referred here as 'without MHW'.

Finally, to match the daily SST time series to EcoTroph-Dyn time-step resolution (fortnight =
1/24 year), we aggregated the initial temperature time series $Y_t$ and 'without MHW' time series at a
fortnight scale. Specifically, we averaged the daily SST for every fortnight of a year. We then computed
the proportion of MHW days (β) within each fortnight. Thus, β = 0 when no MHW day is identified in
a fortnight and β=1 when MHW last for an entire fortnight.





Gridded monthly NPP data from 1998 to 2021 were obtained from satellite-derived estimation. The NPP data were estimated using the EPPLEY-Vertically Generalized Production Model (VGPM)

computation method (Behrenfeld and Falkowski, 1997). The EPPLEY-VGPM method is hybrid model that employs the basic model structure and parameterisation of the standard VGPM computation but replaces the polynomial description of Pb_opt with the exponential relationship described by Morel (1991), based on the curvature of the temperature-dependent growth function described by Eppley (1972).

After excluding spatial cells from the polar biome, we had a total of 34,643 cells, 13,340 of which contained incomplete time series of NPP. For cells with incomplete time series, we utilised the spline function from the R package "stats" to interpolate the missing Net Primary Production (NPP) values. In each ocean cell, the interpolation was constrained by the minimum and maximum satellite data values of the NPP observed over their respective time series. This approach ensured reliable

interpolation and reduced potential bias.

Finally, to match the NPP time series with the resolution of the EcoTroph-Dyn time step (1/24 of a year), we duplicated the monthly NPP values to cover the two fortnights in each month.

### 3.3.2 Biomass simulations

We simulated the changes in biomass spectra in each spatial cell from 1998 to 2021 under the

240 scenarios with and without MHW. We calculated the differences in consumer biomass change between scenarios with and without MHW. Biomass change was computed using the reference period from 1998 to 2009 under the 'without MHW' scenario, allowing us to examine the projected declines in biomass specifically attributable to MHWs. We explored the sensitivity of the results to species' resistance capacity to MHW conditions using the four resistance capacity settings represented by the

245 values of the coefficient α.

We initialised the model by applying a 'burn-in' period of simulating 12 years without any environmental forcing. Indeed, simulation testing indicates that a 'burn-in' period of at least 10 years is needed for biomass spectra to reach equilibrium (S4). Increasing the 'burn-in' period beyond 12 years does not significantly change the equilibrium biomass spectra (t test p-value=0.6118). We thus

used the average of the time series (1998 to 2021) as the 'burn-in' period in each ocean spatial cell.

### 3.3.3 Northeast Pacific Ocean MHW case study

An analysis focusing on the MHW that occurred in the Northeast Pacific Ocean from 2013 to 2016 (commonly known as 'the Blob', Bond et al., 2015) was undertaken to assess the capability of EcoTroph-Dyn to transcribe past MHWs events. First, we extracted the EcoTroph-Dyn simulation

outputs (with and without MHW) in the region, delineated by the boundary of nine biogeochemical provinces that had been identified by Reygondeau et al., (2013) and adapted from Longhurst, (2007).





These biogeochemical provinces include: Central American coast (CAMR), California Ocean and Coastal current (OCAL and CCAL), Alaska coastal downwelling (ALSK), North Pacific Tropical gyre (NPTG), Northeast Pacific subtropical (NPSE), North Pacific polar front (NPPF), Eastern Pacific subarctic gyres (PSAE), and the Western Pacific subarctic gyres (PSAW). Second, we computed biomass change using the reference period from 1998 to 2009 under the 'without MHW' scenario, allowing us to examine the projected declines in biomass specifically attributable to 'the Blob'. Finally, we discussed and compared our results with the literature.

## 4.   Results

### 4.1  Environmental forcing with and without MHWs

In response to climate change, the NPP and SST are already perturbed. Global total NPP decreased by 1% by 2015-2021 compared to the 1998-2009 period. However, large spatial variability in NPP changes were observed (Figure 2a). Notably, NPP decreased by 20% in the Northeast Pacific, while an increase of 20% was estimated for the Southern Ocean. Under the 'without MHW' scenario (Figure 2b), global average SST increased by 0.28°C in 2015-2021 compared to the 1998-2009 period, with some areas warming up to 0.5°C. However, under the 'with MHWs' scenario (Figure 2c), SST increased by 0.32°C in 2015-2021 compared to the period 1998-2009, with some areas, such as the Northeast Pacific, warmed up by 1°C during the same period. Globally, the estimated increase in average SST from 2015 to 2021 was 4% higher in the 'with MHWs' scenario compared to the 'without MHW' scenario.

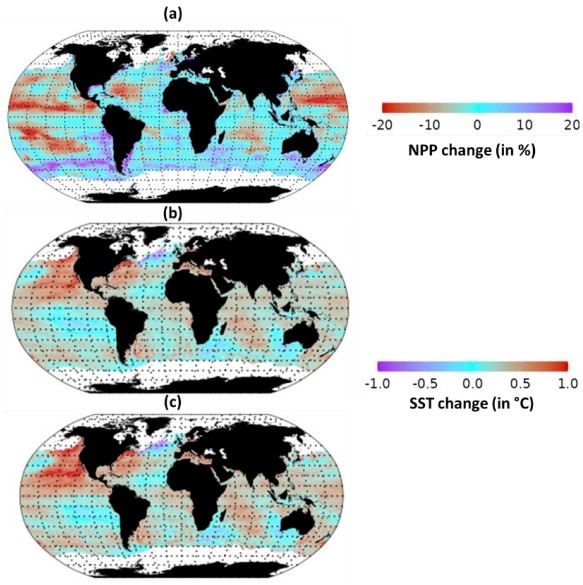



**Figure 2: Changes in SST and NPP between the average of 2015-2021 relative to the average between 1998-2009.** (a) NPP, (b) SST under the 'without MHW' scenario, and (c) SST under the 'with MHWs' scenario.

Under the 'with MHWs' scenario, MHWs occurring during the year's warmest month increased in intensity, duration, and surface extent from 1998 to 2021 (Figures 3a, b) with large spatial variability (Figures 3c, d). MHWs with intensity lower than 3°C above the climatology were identified on average in 28.5 % of the ocean surface. These MHWs lasted, on average, more than 40 days. In contrast, MHWs characterised as higher intensity (≥3°C above climatology) were identified in <20% of the ocean

surface area. These relatively more intensive MHWs lasted, on average, 32 days. Furthermore, more days with MHWs with lower intensity were identified for low latitude regions (23°N - 6°S) (Figure 3c) compared to MHWs identified in higher latitude regions (> 23°N and 25°S). In addition, the intensity of MHWs was higher in higher latitude regions in the northern hemisphere relative to those in the southern hemisphere (Figure 3d).

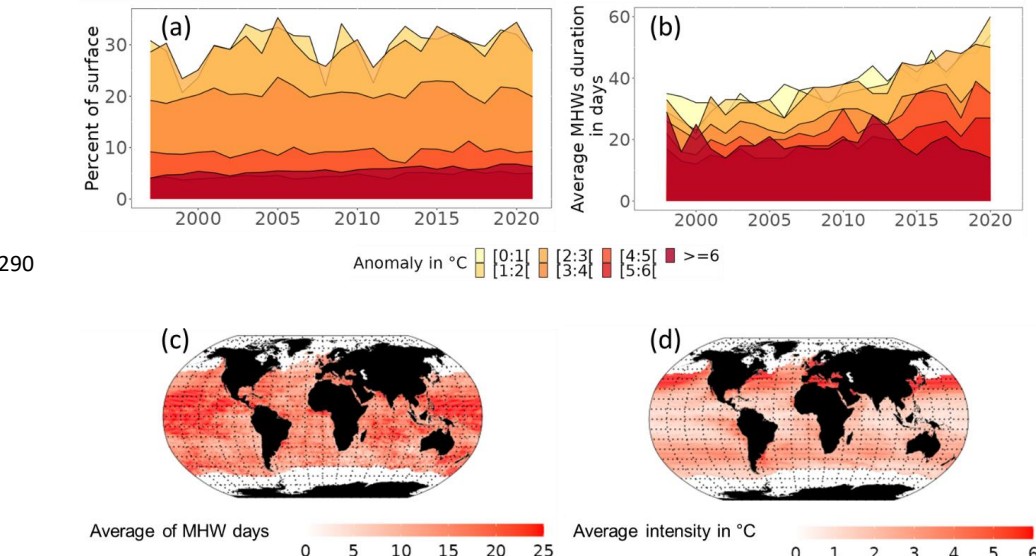


**Figure 3: Temporal and spatial characteristics of MHWs identified for the period 1998 to 2021.** (a) Evolution of spatial extent of MHWs categorised by their intensity, (b) Evolution of MHWs averaged duration categorised by their intensity, and (c) Average intensity of MHWs over the period 2015-2021. (d) Average number of MHWs over the period 2015-2021.

## 4.2   Biological impacts of MHWs at the global scale

### 4.2.1 Impact on total consumer biomass

        The projected changes in total consumer biomass were higher under the 'with MHWs' scenario relative to the 'without MHW' scenario.  Globally, total consumer biomass was projected to decrease, on average, by 0.07 ± 0.02% per year (standard error) relative to the baseline period of 1998-2009

(Figure 4a) (GLS, Generalized Least Squares, p-value < 0.05). In contrast, simulations under the 'with





MHWs' scenario with an MHW resistance capacity setting of α=0.2 projected an average decrease in total consumer biomass of 0.12 ± 0.02% per year relative to the baseline period of 1998-2009 (p-value < 0.05). Thus, simulations that focused on annual mean changes in temperature only masked the effects of the short-term impacts of MHWs on the long-term changes in consumer biomass in the

ecosystem (S5).


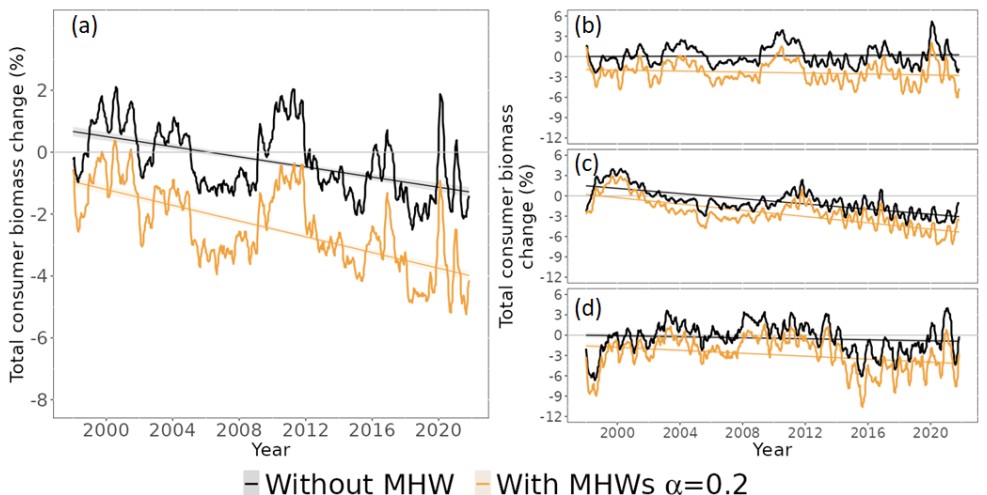

**Figure 4:  Simulated total consumer biomass change in the ocean from 1998 to 2021 under the**
**'without MHW' (black lines) and with MHWs (light orange lines) scenarios, with a resistance capacity**
**setting of α=0.2.** (a) the world ocean (excluding polar biome), (b) temperate biomes, (c) tropical biomes, and (d) upwelling biomes, respectively (S2 for biomes spatial definition). The horizontal grey line separates positive and negative total consumer biomass changes.

While total consumer biomass was projected to decrease slightly across the three biomes (Figures 4b, c, and d), in the scenario 'with MHWs' the declines in tropical biomes were larger than the global average. Under the "without MHW" scenario, total consumer biomass in temperate biomes was projected to increase by 0.01% ± 0.01 (standard error), while in tropical and upwelling biomes, it was projected to decrease by 0.18% ± 0.01, and 0.02% ± 0.01 per year, respectively, relative to the baseline





period. Under the 'with MHWs' scenario with α=0.2, total consumer biomass in temperate, tropical, and upwelling biomes was projected to decrease more strongly by 0.03% ± 0.01%, 0.23% ± 0.02%, and 0.1% ± 0.02, respectively, relative to the baseline.

### 4.2.2 Impacts by trophic level

MHWs exacerbated the projected climate change impacts on biomass, particularly for higher

trophic level groups. Indeed, our model projected a similar level of biomass loss across trophic levels in the 'without MHW' scenario, with a projected decrease in global total consumer biomass by 1 ± 0.1% SE by 2015-2021 relative to 1998-2009 (Figure 5a). In contrast, under the 'with MHWs' scenario and with a resistance capacity setting of α=0.2, total consumer biomass was projected to decrease by 4.4 ± 0.1% for high trophic level classes (TL ∈ [4;5.5]), while the decrease was smaller for low trophic

level (TL ∈ [2;3[, 3.4 ± 0.1%) and mid trophic level (TL ∈ [3;4[, 4.1 ± 0.1%) classes (Figure 5e).

The impact on high trophic levels differed between biomes, notably with the tropical and upwelling biomes being more impacted. Under the 'without MHW' scenario, our model projected a decrease in high trophic level consumer biomass of 0.5 ± 0.1%SE, 2.4 ± 0.1%, and 2.2 ± 0.1% in temperate, tropical, and upwelling biomes, respectively, by 2015-2021 relative to 1998-2009 (Figures

5b, c, and d). In contrast, under the 'with MHWs' scenario and with a resistance capacity setting of α=0.2, high trophic level consumer biomass was projected to decrease by 3.3 ± 0.1%, 6.6 ± 0.1%, and 5.9 ± 0.1% in temperate, tropical, and upwelling biomes, respectively, while a smaller biomass decrease for low and mid-trophic level was expected (Figures 5e, g, and h).

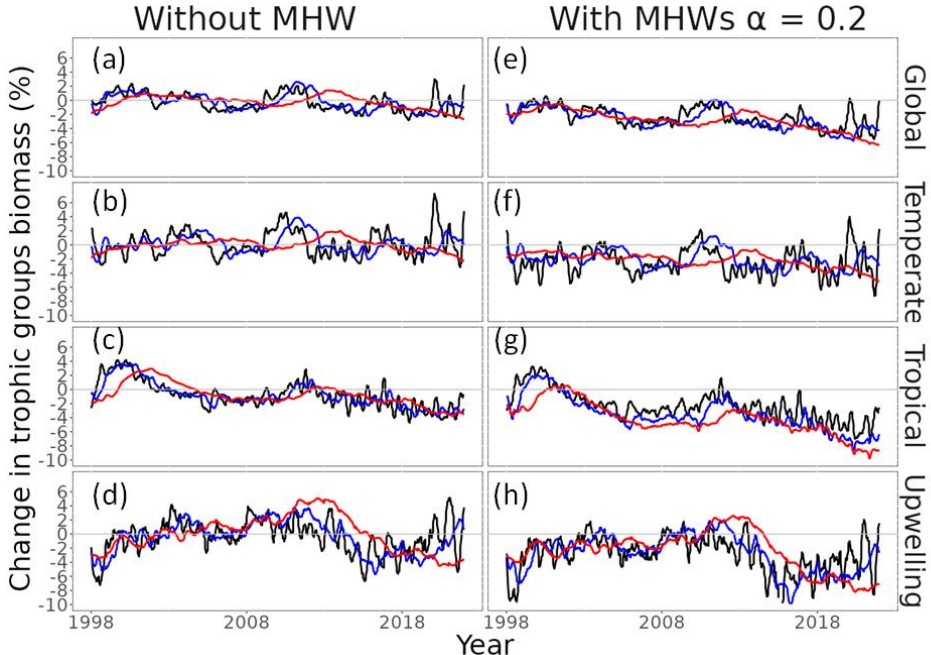



**Figure 5: Projected changes in consumer biomass by trophic levels and biomes under 'without MHW' and 'with MHWs' scenarios relative to the average between 1998-2009.** Low trophic level (TL ∈ [2;3[ in black), medium TL (TL ∈ [3;4[ in blue), and high TL (TL ∈ [4;5.5] in red). (a, e) correspond to global scale, (b, f) to the temperate biome, (c, g) to the tropical biome, and (d, h) to the upwelling biomes. (a, b, c, and d) correspond to 'without MHW' scenarios while (e, f, g, and h) correspond to 'with MHWs'
scenarios. The simulation results under the 'with MHWs' scenario used a resistance capacity setting of α=0.2.

### 4.2.2 Various responses of ecosystems to MHWs

Over the 2015 to 2021 period, MHWs impacted total consumer biomass with large variations in both the direction and magnitude of the impacts between biomes (Figure 6). Under the 'without

MHW' scenario, the model projected an increase in total consumer biomass in 33.5% of the ocean area, especially in temperate biomes (Figure 6a). Under the 'with MHWs' scenario, this proportion decreased and a biomass decrease was projected to occur in 76% of the global ocean, with a projected global biomass decrease of 3.6% by 2015-2021 relative to 1998-2009. In high latitudes (> 25°N and 25°S), MHWs exhibiting high intensity but short duration (Figures 3c and d) resulted in moderate

additional biomass losses, up to 8% in 2015-2021 (Figure 6c). Conversely, in low latitudes (25°N - 25°S), MHWs led to substantial additional biomass losses, exceeding 10% on average (Figure 6c) and up to 20% in biodiversity hotspots such as Indonesia, off Papua New Guinea Coast, and Central America. Thus, high latitude and upwelling areas appeared to be refuge zones from MHWs compared to low latitudes. Looking at the MHW's additional impact within food webs (S6, S7, and S8), similar spatial

patterns and responses were expected with a higher impact (biomass decreases) on high trophic levels compared to lower trophic levels.

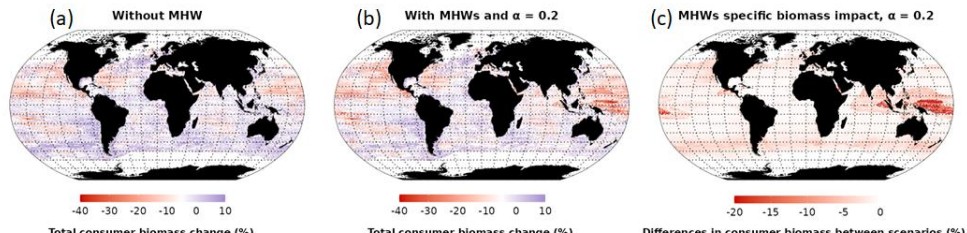

**Figure 6: Changes in total consumer biomass in 2015-2021 compared to 1998-2009.** (a) total consumer biomass under the "without MHW" scenario, (b) total consumer biomass under the 'with
MHWs' scenario with α=0.2, and (c) differences in consumer biomass between scenarios.

## 4.3   Sensitivity to resistance capacity setting (α)

Ecosystem total consumer biomass response to MHWs depended on community resistance capacity. Assuming species to have full resistance (α=0) or no resistance (α=1) to MHWs, total
consumer biomass was projected to decrease by 2.3% and 15.3% by 2015-2021 relative to 1998-2009, respectively (Figure 7a). Simulations using different α values projected changes in total consumer biomass that were consistent in direction but differed significantly in magnitude. A three-way ANOVA



was performed to analyse the effect of trophic levels, biomes, and α value on biomass change. This
ANOVA revealed a statistically significant effect on biomass change of trophic levels, biomes, and α

values (p <2e-16), individually. Additionally, the interaction between biomes, trophic levels, and α
value also had a significant effect on biomass change (F (27, 16737) = 442.4, p-value < 2e-16). The
lower the resistance capacity of species (higher α value), the greater the loss of total consumer
biomass. On a biome scale, these variations in biomass loss were greater for the tropics, with values
ranging from 3.1% (α=0) to 20.2% (α=1) (Figure 7c). In comparison, sensitivity was lower in the

temperate and upwelling biomes with a decrease between 1.4% (α=0) and 11.2% (α=1) and 3.8% (α=0)
and 12.3% (α=1), respectively (Figures 7b and d).

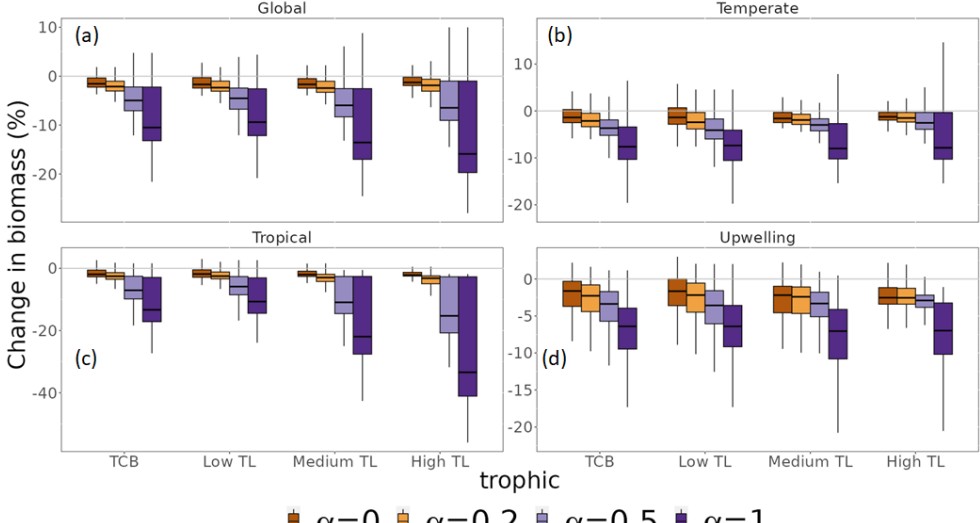

**Figure 7: Sensitivity of changes in total consumer biomass to different resistance capacity settings.
Changes are aggregated for the whole biomass spectrum (TCB) and by trophic levels (TL) (low TL,**
**medium TL and high TL).** Colours represent different resistance capacity settings (α values). (a) Global
ocean, (b) temperate biomes, (c) tropical biomes, and (d) upwelling biomes. The different horizontal
lines of the box plot refer to the median, 25th, and 75 quantiles.

The sensitivity of changes in total consumer biomass to the resistance setting increased with
trophic level. At a global scale, low TLs response ranged from a consumer biomass decrease of 1.4%
(α=0) to 8.0% (α=1), while high TLs exhibited values ranging from 0% (α=0) to 11% (α=1) (Figure 7a).
Furthermore, the sensitivity of trophic levels to the α value was greater in the tropical biome and lower
in the temperate and upwelling biomes relative to that of the global ocean.





### 4.4 A case study of a Northeast Pacific MHW

Our model projected that the MHW in the Northeast Pacific from 2013 - 2016 ('the Blob') resulted in long-term changes in biomass spectrum in the region (Figure 8a). Temperature anomalies were >=4 °C (July-October 2013 to 2016 average) and up to 8 °C in 2015 relative to 1982-2011. This temperature anomaly had some ecological repercussions. On average, without accounting for MHWs,

total consumer biomass in our simulation was hindcast to decrease by 1.5% ± 0.3, 3.1% ± 0.4, 6.6% ± 0.2, and 5.2% ± 0.3 in 2013, 2014, 2015, and 2016, respectively, compared to the reference period of 1998 to 2009 (black line in Figure 8b). However, when accounting for MHWs with α=0.2, the total consumer biomass loss decreased on average by an additional 3.1% in 2013, 2014, 2015, and 2016. Furthermore, the difference in linear trend (slope and intercept) between the biomass time series 1998

to 2013 and 2017 to 2021 indicated that the MHW ('the Blob') had a significant effect on long-term changes in biomass (GLS, Generalized least squares, p_value < 0.05).

      The magnitude of the hindcast total consumer biomass differed significantly between biogeochemical provinces (ANOVA, $F_{(1, 2906)}$ = 7.854, p-value < 0.05, Figure 8c). Comparing the consumer biomass before (1998 to 2012) and during (2013 to 2016) the occurrence of 'the Blob', all

biogeochemical provinces in the Northeast Pacific, except Alaska coastal downwelling (ALSK) and Central American coast (CAMR), exhibited significant total consumer biomass decrease (ANOVA, $F_{(1, 151)}$ = 155.443, p-value < 2e-16) under the scenario with and without MHW. Under the 'without MHW' scenario, total consumer biomass in the North Pacific Tropical gyre (NPTG) biogeochemical provinces, followed by the California current (CCAL+OCAL biogeochemical provinces), was hindcast to decline

most amongst the provinces when comparing before, during, and after 'the Blob' (Figure 8c). However, under the 'with MHWs' scenario and with a resistance capacity setting of α=0.2, all biogeochemical provinces were hindcast to have an additional biomass decrease during and after 'the Blob'. This additional biomass decrease ranged from 0.9% to 5% and 1.4% to 5.1% during and after 'the Blob', with an average decrease of 2.7% ±0.4% and 3.1% ± 0.3%, respectively (Figure 8c). Particularly, the

California current and the Alaska coastal downwelling provinces were the most affected by MHW, with additional biomass decreases of 5% and 3.8% 'with MHWs' relative to 'without MHW' scenario, during and after 'the Blob'.

      The impact of 'the Blob' on various trophic levels was similar to the global pattern, with a rapid reaction (in terms of biomass loss) observed at lower TL and higher TLs exhibiting a delayed response

(S9). When considering the influence of MHWs from 2013 to 2016 and under the 'with MHWs' and α=0.2 resistance capacity, there was a hindcast biomass reduction of 6.8% ± 0.7, 6.3% ± 0.6, and 4.3% ± 0.3 for low, medium, and high TLs, respectively, when compared to the reference period of 1998 to





2009. Compared to pre-event conditions, as of 2021, aside from low TLs, medium and high TLs gave no sign of recovery from 'the Blob' MHW event.


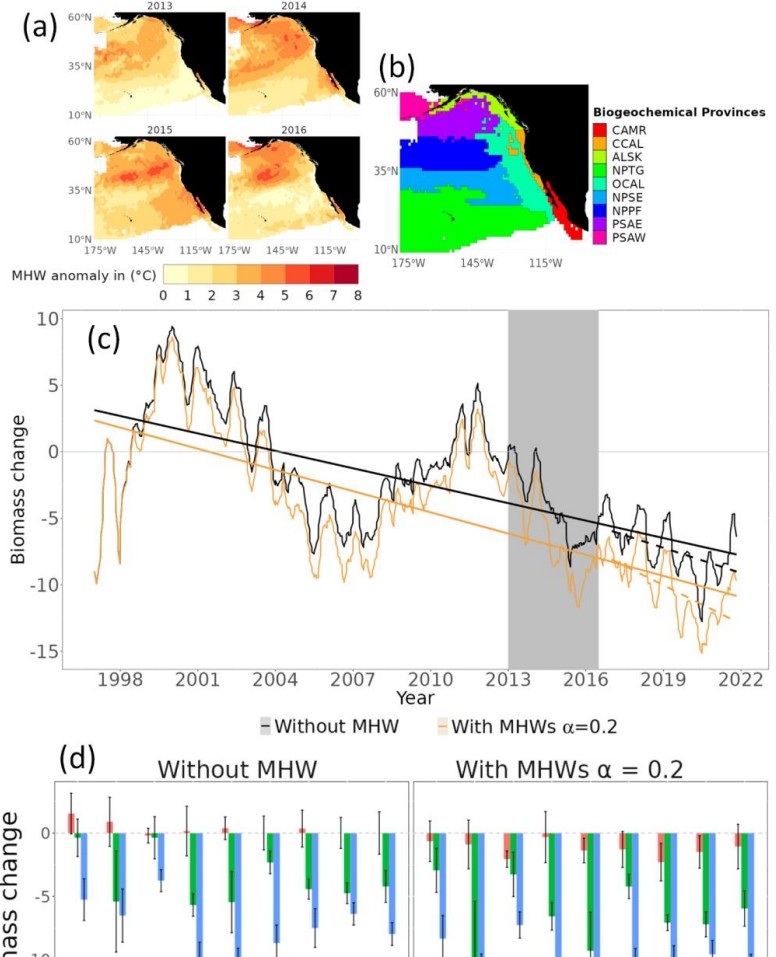

**Figure 8: Hindcast of temporal and spatial biological impacts of the MHW from 2013 to 2016 in the Northeast Pacific ('the Blob').** (a) Temperature anomalies between 2013 and 2016, (b) biogeochemical provinces definition, with Central American coast (CAMR), California Ocean and Coastal current (OCAL and CCAL), Alaska coastal downwelling (ALSK), North Pacific Tropical gyre (NPTG), Northeast Pacific subtropical (NPSE), North Pacific polar front (NPPF), Eastern Pacific subarctic gyres (PSAE), and the Western Pacific subarctic gyres (PSAW). (c) Generalised least squares models (lines) and total consumer biomass change 'without MHW' and 'with MHWs' scenarios in the black and light orange colour, respectively. The vertical grey shaded area indicates 'the Blob' duration. (d) Average total




consumer biomass changes before (1998 to 2012, pink), during (2013 to 2016, green), and after (2017
        to 2021, blue) 'the Blob' for each biogeochemical province. The error bars correspond to standard
        errors.

        Trophic dynamics of the biomass spectra in the Northeast Pacific were hindcast to be impacted
        by 'the Blob' (S10). The trophodynamic indicators flow kinetic and trophic efficiency increased and

decreased, respectively, in the model. Under the scenario 'without MHW', the average hindcast flow
        kinetic of the biomass spectra in the Northeast Pacific increased by 0.6% ± 0.2 while the transfer
        efficiency decreased by 0.7% ± 0.2 by 2013-2016 relative to 1998-2009. Consideration of the MHW
        under the 'with MHWs' scenario exacerbated the increase in flow kinetic and decrease in trophic
        efficiency significantly by 0.7% and 0.4%, respectively (t test p-value < 0.05).

## 5. Discussion


        In this study, by accounting for MHWs in the last four decades' simulation hindcast, we showed
        the potential of synergic impacts of MHWs (pulses) with the trend in temperature induced by long-
        term directions of climate change (presses) on biomass and trophodynamics of ecosystems (Bender et
        al., 1984; Harris et al., 2018). These reconstructed MHW impacts varied spatially because of MHW

spatial dynamics, regional differences in ocean biogeochemical and physical conditions and ecosystem
        trophodynamic characteristics.  Furthermore, studying 'the Blob', we highlighted that MHWs could
        exacerbate the long-term impacts of climate change that vary between biogeochemical provinces.

### 5.1  Impacts of MHWs on global marine consumer biomass

        Over the past two decades, MHWs have led to increased mortality rates and significant

alterations in the functioning and structure of ecosystems (Smith et al., 2023; Wernberg et al., 2013,
        2016). Our modelling analysis suggested that, without accounting for MHWs and their ecological
        effects, the decline in global-scale biomass associated with climate change may have been significantly
        underestimated. We showed that MHWs exacerbated the impacts of long-term climate change,
        impacting trophodynamic parameters such as the flow kinetic and transfer efficiency within

ecosystems congruent with Arimitsu et al., 2021; Gomes et al., 2024; Smith et al., 2023 studies. These
        perturbations of ecosystems functioning result in biomass loss through food webs. Although MHW
        temperature anomalies have mostly lasted for weeks to months only, we emphasise that the resulting
        ecosystems perturbations last for a longer time (decades) and influence long-term biomass change
        (Figure 8, Babcock et al., 2019; Cheung & Frölicher, 2020; Guibourd de Luzinais et al., 2024). Finally,

the intensity and duration of MHWs influenced the magnitude of ecosystem functioning perturbation
        (Oliver et al., 2021; Smith et al., 2023). We highlighted that the intensity and duration of MHWs
        characteristics have continuously increased since the beginning of the 21st century (Figure 3), leading
        to a sharp biomass decrease over the period hindcast (Figure 4). Such short-term biomass decreases



strengthened the impacts of long-term climate change (Cheung and Frölicher, 2020; Collins et al.,
2019).

At a global scale, over the whole time series and different MHW resistance capacity scenarios,
high TL biomass experienced greater impacts from MHWs that were not able to recover pre-perturbed
levels as much as the low and medium TL biomass. The gap in biomass recovery was most apparent in
tropical and upwelling biomes where the hindcast biomass of high TL consistently decreased over time.
The time needed for the biomass to return to pre-perturbed levels was related to the rate of biomass
turnover, which was also dependent on the speed of biomass flows. This turnover decreased with
trophic level and was low for organisms at the top of the food chain (Gascuel et al., 2008; du Pontavice
et al., 2020; Schoener, 1983). Thus, the high frequency of MHWs in recent years may, therefore, be
greater than the time that high TLs need to recover from the impacts of individual MHWs and could
contribute to the continuing decline in high-TL biomass.

## 5.2 Trophodynamics approach as a framework for deciphering ecosystem responses to MHWs

In our modelling approach, trophodynamic changes were assessed through transfer efficiency
(TE) and flow kinetics. TE (dimensionless) is defined as the fraction of energy transferred from one
trophic level (TL) to the next and summarises all the losses in the trophic network (Jennings et al., 2002;
Libralato et al., 2008; Lindeman, 1942; Niquil et al., 2014; Pauly and Christensen, 1995; Schramski et
al., 2015; Stock et al., 2017). It is an emergent property of marine ecosystems and an essential
parameter in many applications of marine ecology, such as estimating biomass flux in production
models (e.g., Carozza et al., 2016; du Pontavice et al., 2021; Gascuel et al., 2011; Jennings et al., 2008;
Tremblay-Boyer et al., 2011). Unlike previous applications of EcoTroph at an annual timescale (du
Pontavice et al., 2021, 2023), where changes in transfer efficiency reflected long-term changes in
species assemblages, having a transfer efficiency that evolved on a biweekly basis (14 days) allowed us
to take into account metabolic fluctuations over a shorter period. The temperature during MHWs
increases the basal metabolism (catabolism) of marine organisms (Grimmelpont et al., 2023; Minuti et
al., 2021), thus reducing the energy available for anabolic processes that can only occur once catabolic
needs are met (Eddy et al., 2021). The estimation of transfer efficiency only considers anabolic
processes (Eddy et al., 2021); hence, this balance between catabolism and anabolism is how MHWs
could disrupt transfer efficiency on a biweekly scale. The flow kinetics expressed in TL·year$^{-1}$ quantifies
the speed of the trophic flux, i.e., the rate of biomass transfers from lower trophic levels to higher
ones, due to predation and/or ontogenesis (Gascuel et al., 2008; du Pontavice et al., 2020). This rate
is inversely proportional to the biomass residence time (BRT, du Pontavice et al., 2020), which is the
average time a unit of biomass spends moving from one TL to the next higher one through predation



(Gascuel et al., 2008; Schramski et al., 2015). This flow kinetics can be measured by the production/biomass ratio (P/B, Gascuel et al., 2008), used in many ecosystem models, particularly EwE (Ecopath with Ecosim, Christensen & Pauly, 1992).

MHWs disrupt the metabolism of individuals and, on a larger scale, impact the entire ecosystem with trophodynamic changes throughout the trophic network (S10, Arimitsu et al., 2021; Collins et al., 2019; Gomes et al., 2024; Smith et al., 2023). However, due to their different physical characteristics depending on the ecosystems and the varying functioning of these ecosystems, they do not induce the same structural and functional changes, leading to different biomass losses. Tropical ecosystems are composed of species with low metabolic efficiency (TE) (low ratio between ingested energy and stored energy), with significant energy losses that increase with temperature (Brown et al., 2004; du Pontavice et al., 2020; Schramski et al., 2015). To compensate for this low efficiency, predation activity is high, resulting in rapid biomass transfers between prey and predators (biomass flow, du Pontavice et al., 2020). In these ecosystems, communities (which may be dominated by short-lived and fast-growing species) are generally living at the upper end of their thermal preferences (Begon and Townsend, 2021; Pinsky et al., 2019; Vinagre et al., 2016). They experience thermal stress associated with MHWs, leading to substantial mortality across the trophic network, with a higher proportion occurring in lower trophic levels regardless of the intensity of the MHWs (Guibourd de Luzinais et al., 2024). MHWs scarcely affect transfer efficiency (TE), with an estimated average decrease of 0.05% between 1998 and 2021 because the metabolism of species is already very high (du Pontavice et al., 2020). However, MHWs exacerbate prey mortality rates, resulting in proportionally higher predation rates; i.e., biomass remains for a shorter time at each trophic level, corresponding to a sharp acceleration of biomass flow (estimated at 1 % on average between 1998 and 2021). Given the relatively high number of MHW days (Figure 3, Hobday et al., 2018; Marin et al., 2021; Oliver et al., 2018), these modifications to biomass flow are "persistent" and involve significant biomass losses.

In temperate ecosystems, biomass is transferred more slowly between trophic levels with less loss (higher trophic efficiency, du Pontavice et al., 2020; Eddy et al., 2021). In EcoTroph-Dyn, temperate communities experience thermal stress and mortality only during high-intensity MHWs (Guibourd de Luzinais et al., 2024). MHWs affect the metabolic efficiency of species by increasing basal metabolism and respiration-related losses, which decreases TE across the trophic spectrum, estimated in our simulations at an average of 0.2% between 1998 and 2021. These higher metabolic demands lead to an increase in predation activity, which, combined with lower mortality (compared to tropical waters), reduces the biomass residence time at each trophic level in the food chain, i.e., a slight acceleration of the speed of biomass flow between each trophic level (estimated in our simulations at an average of 0.3% between 1998 and 2021). Since MHWs in these ecosystems are characterised by significant



temperature anomalies but over relatively short durations (Figure 3, Frölicher et al., 2018; Oliver et al., 2018), these modifications to biomass flow are "temporary" and involve lower biomass losses.

Finally, in upwelling ecosystems, communities are characterised by low transfer efficiencies
with high biomass residence times, indicating slow biomass transfers between prey and predators (du Pontavice et al., 2020). These MHWs cause thermal stress and moderate mortality (Guibourd de Luzinais et al., 2024). They lead to a decrease in the metabolic efficiency of species (estimated in our simulations at an average of 0.03% between 1998 and 2021), along with an increase in predation activity, moderately impacting the rate of biomass flows within the ecosystem (estimated in our
simulations at an average of 0.3% between 1998 and 2021). The MHWs occurring in these ecosystems are of high intensity and last relatively long (an average of 23 days per year between 1998 and 2021). One might expect more significant impacts than in tropical environments; however, this was not the case. This could potentially be explained by (i) the high productivity of these ecosystems, which supports a highly biodiverse food web (Largier, 2020; Pauly and Christensen, 1995; Rykaczewski and
Checkley, 2008; Ryther, 1969), leading to better resilience to environmental changes and extreme temperature events (Bernhardt and Leslie, 2013), and (ii) the specific functioning of these ecosystems, which tends to reduce the number of MHW days compared to their adjacent open ocean (Varela et al., 2021).

Generally, through our modelling approach, higher temperatures and frequent mortality
events promoted the emergence of species and the growth of populations adapted to warm waters, both characterised by rapid growth and short lifespans (Beukhof et al., 2019; du Pontavice et al., 2020). This phenomenon was observed, for example, during 'the Blob' event in California Ocean and Coastal Current (OCAL and CCAL) provinces, with the increase of tropical species that usually live much further south, such as tuna, sailfish, and marlin (Cavole et al., 2016).

The specific examination of 'the Blob' allowed us to elucidate how this MHW differently affected the biogeochemical provinces (Longhurst, 2007; Reygondeau et al., 2013) and influenced their long-term responses. According to our simulations (Figure 8), the biomass in the affected oceanic regions had not returned to their reference levels (1998-2009) by 2021. Following the MHW, all biogeochemical regions experienced significant biomass losses (ANOVA, p_value < 0.05), though with
varying magnitudes. Differences in exposure to the intensity and duration of temperature anomalies can potentially explain these differences in responses. For example, the coastal part of the California Current and the subarctic gyres of the North Eastern Pacific were subjected to lower anomalies over different durations (Varela et al., 2021). In addition, differences in the structure and functioning of trophic networks may also have contributed to the variation in responses to MHWs between provinces
(Morgan et al., 2019; Peterson et al., 2017; Ruzicka et al., 2012). The biogeochemical zones have different compositions of pelagic communities that may respond differently to MHWs (Peterson et al.,



2017). For instance, the Gulf of Alaska saw its planktonic community shift towards smaller plankton and zooplankton, resulting in a loss in the nutritional quality and quantity of the forage portion of the pelagic community for their predators, highlighting the disruption of biomass flow (Arimitsu et al.,

2021; Piatt et al., 2020; Suryan et al., 2021). In contrast, in the California Current biogeochemical region, the MHW was associated with, a substantial increase in the abundance of pyrosomes limiting/stopping energy flow moving toward higher trophic levels has been observed (Gomes et al., 2024). This was not necessarily the case for the oceanic part of the California Current. These different changes in the composition and abundance of the lower trophic levels are represented in EcoTroph-

Dyn by changes in the amount of energy flowing through the food web. EcoTroph-Dyn also takes into account the differences in trophodynamic characteristics (e.g., transfer efficiency, flow kinetics) between biogeochemical zones. For example, communities in the Alaskan Gulf are more efficient than those in the Californian Current (du Pontavice et al., 2020) and the energy entering the food web was less disrupted than in the Californian Current, which may explain the greater impact of the MHW on

the Californian Current.

## 5.3   Model validation and sources of uncertainties

We have developed an innovative approach to modelling marine ecosystems, linking trophic ecology with MHWs hindcast to assess their ecological impacts on a seasonal time scale and at a global scale. Despite its apparent simplicity and the reduced number of parameters, EcoTroph-Dyn is part of

the family of "complete ecosystem models and dynamic system models" (Plagányi, 2007). This model accounts for all trophic levels, from primary producers to top predators. It merges individual "species" into categories defined solely by their trophic levels and describes ecosystems through a continuous distribution of biomass (trophic biomass spectrum), from primary producers to top predators. EcoTroph-Dyn does not account for specific climate effects on individual species and populations. The

model assumes that variations in environmental conditions will lead to new biomass transfer dynamics in theoretical ecosystems at equilibrium. EcoTroph-Dyn meets the criteria described by Link, (2010) to be considered a plausible representation of marine ecosystems. These criteria include: i) the biomass density values of all functional groups must cover 5 to 7 orders of magnitude, ii) there must be a 5 to 10% decrease in biomass density (on a logarithmic scale) for each unit increase in trophic level, iii)

specific biomass production values (P/B) must never exceed specific biomass consumption values (C/B), and iv) the ecotrophic efficiency (EE) for each group must be less than 1. Additionally, the model relies on empirically obtained equations (Gascuel et al., 2008; du Pontavice et al., 2020) and has demonstrated its ability to replicate the effects of fishing and climate change on marine ecosystems (e.g., du Pontavice et al., 2021, 2023; Tremblay-Boyer et al., 2011).



To discuss the validity of our hindcasts, we focussed on the most studied MHW ('the Blob') in the literature with available quantitative data (e.g., Arimitsu et al., 2021; Bond et al., 2015; Cavole et al., 2016; Cheung & Frölicher, 2020; Gomes et al., 2024; Piatt et al., 2020; Suryan et al., 2021). The projected biomass changes from our 'with MHWs' simulations and the α=0.2 scenarios showed the same biomass evolution per trophic group as the observational data based on species biomass surveys

(Suryan et al., 2021). However, our projections estimated lower biomass and also showed a smaller biomass decrease compared to observed data (Suryan et al., 2021). This would indicate a possible underestimation of biomass and ecosystem response to MHWs by EcoTroph-Dyn.

Our simulations of biomass changes, according to the 'with MHWs' and α=0.2 scenarios, also fell within the range of biomass projections reported by studies using other approaches to represent

ecosystem functioning (e.g., species-based Dynamic Bioclimate Envelope Model (DBEM) by Cheung & Frölicher, 2020, and Ecotran, an extension of the popular Ecopath modelling framework by Gomes et al., 2024, see Figure 9). Additionally, we projected that the trophodynamic kinetic parameter of biomass flow increased by 3% in the 'with MHWs' and α=0.2 scenarios, which is consistent with the estimates highlighted by Gomes et al., 2024 (3.7%). Finally, the decreases in trophodynamic transfer

efficiency parameters highlighted in EcoTroph-Dyn with 'the Blob' case study correspond to previous estimates (e.g., Arimitsu et al., 2021) showing a reduction in the availability/quality of food in the food web. It is worth noting that projections obtained from a smaller (larger) α led to an underestimation (overestimation) of biomass losses and changes in biomass flow parameters relative to Cheung & Frölicher, 2020 and Gomes et al., 2024 estimates.




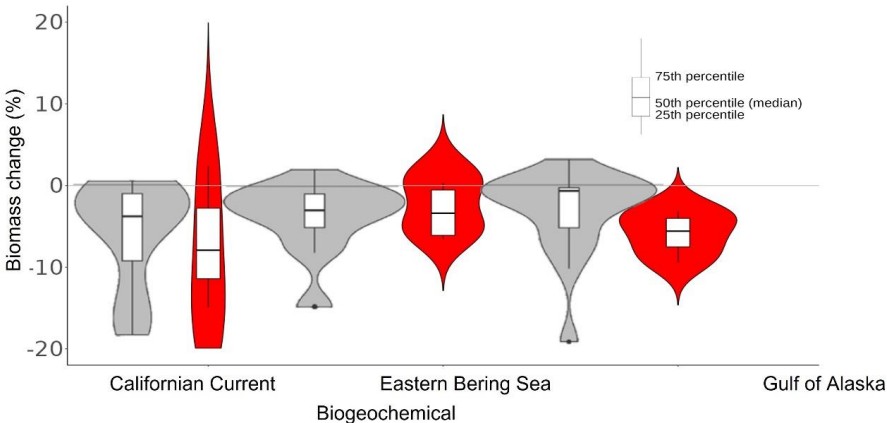

**Figure 9: Biomass change associated with the Northeast Pacific MHW (2013-2016) per**
**biogeochemical province.** Grey violin plots correspond to results from Cheung et al., (2020), while red
corresponds to our hindcasted EcoTroph-Dyn simulation with α=0.2 scenario. (Figure adapted from
Cheung & Frölicher, 2020).

One of the main uncertainties in modelling MHWs in EcoTroph-Dyn is the assumption of the
resilience of the biomass spectrum to MHWs, i.e., parameter α. It is important to note that the α values
applied in this study were chosen arbitrarily, albeit reasonably, and were able to capture a broad range
of potential responses to MHWs. It would thus have been valuable to apply the same process of
comparison with other MHWs in the world ocean in order to acquire better estimates of the α
parameter and more reliable results regarding the consequences of MHWs. However, to date, few
MHWs are as well studied as 'the Blob', limiting these analyses. As discussed in Guibourd de Luzinais
et al., 2024, given the diversity of ecosystem functioning, there is no reason why communities in
different ecosystems, biogeochemical regions or trophic levels should have the same capacity to resist
MHWs. Thus, we encourage future studies to use various observational data on the impacts of MHWs
across the ocean (if available) to better estimate the α parameter as a function of different ecosystems,
biogeochemical regions or trophic levels in order to reduce the uncertainties in projections.

Furthermore, uncertainties about our results arise from EcoTroph-Dyn environmental drivers.
In our current investigation, EcoTroph-Dyn has been driven by satellite data. For the NPP forcing
variable, we implemented the EPPLEY-VGMP algorithm (Behrenfeld and Falkowski, 1997; Morel, 1991),
which, like other algorithms such as CbPM and CAFE, derives NPP from satellite-derived estimates of
Chl-a and SST. In this study we did not consider the 'with' and 'without' MHWs scenario for NPP.
Acknowledging that MHWs generally increase NPP at high latitudes while decreasing it at low latitudes
(Arteaga and Rousseaux, 2023; Bouchard et al., 2017; Le Grix et al., 2022; LeBlanc et al., 2020), we
could have overestimated MHWs impacts on ecosystem functioning at high latitudes and
underestimated their impact at low latitudes. Furthermore, the variability and uncertainty in the



estimation of the NPP by satellites directly affects the reliability of our results. In EcoTroph-Dyn, as well as in other Marine Ecosystem Models, ocean primary production (and its related phytoplankton biomass) plays a crucial role in both sustaining and constraining the biomass of higher trophic levels (e.g., Blanchard et al., 2012; Carozza et al., 2016; Cheung et al., 2011; Jennings & Collingridge, 2015). However, there exists significant variability in NPP estimation among satellite NPP algorithms

(Milutinović and Bertino, 2011; Westberry et al., 2023). These discrepancies are particularly pronounced over continental shelves and oligotrophic gyres, primarily due to variations in model parametrisation and growth rate representation (Milutinović and Bertino, 2011; Westberry et al., 2023). Consequently, elucidating the sources of the current uncertainty associated with satellite-derived NPP and refining these estimates pose significant challenges in comprehending the responses

of marine food webs to MHWs.

Moreover, EcoTroph-Dyn does not account explicitly for species; thus, we could only assess aggregated food web responses to MHWs. The model projected that the aggregated response across species is expected to be negative, although it is known that some species would exhibit positive effects from MHWs occurrence (Cavole et al., 2016; Smith et al., 2023; Suryan et al., 2021; Wernberg

et al., 2016). To be cautious, we considered various loss rate scenarios to obtain a possible range of responses from marine ecosystems and dismiss any possibility. Running these five MHW loss rate-induced scenarios, we found that the aggregated resistance capacities α had a significant (ANOVA, p_value<2e-16) and large effect. All scenarios are consistent with each other, with a worsening biomass loss over marine ecosystems with resistance capacities decreasing (α increasing). Even

though the global trend of MHWs impacts is negative, to improve and better understand MHWs ecological impacts, we believe that studies using species explicit-based modelling could investigate how various impacts of climate change and how species-level responses will affect trophodynamics and ecosystem structure and functions.

## 5.4  Implications and future research

From our study, we highlighted the specific impacts of MHWs on ecosystem structure and function, particularly through the case study of the MHW known as 'the Blob' (the longest and strongest period of abnormal temperature ever recorded (Collins et al., 2019; Oliver et al., 2021)). The anomalous low wind during the 2013-2014 winter induced anomalously weak Ekman transports of colder water from the north and, coupled with anomalous low air-sea heat exchanges, triggered 'the

Blob' (Bond et al., 2015). Furthermore, these processes as well as the El Niño Southern Oscillation (ENSO) already contributed to the increased average duration and intensity of MHWs in the North-eastern Pacific Ocean. Given that projections for the 21st century indicate that ENSO events will increase in intensity and frequency (Holbrook, Gupta, et al., 2020; Oliver et al., 2021), events such as



'the Blob' are expected to occur more frequently (Holbrook, Gupta, et al., 2020; Oliver et al., 2021).
This underscores the need for ongoing research to understand better how MHWs disrupt ecosystems.
Finally, throughout the 21st century, ecosystem responses will depend on the ability of communities
to adapt to the long-term increase in ocean temperature and their ability to withstand short-term
extreme temperatures (Johansen et al., 2021). Therefore, to enhance our understanding of how
marine ecosystems will respond to climate change, future studies should focus on potential scenarios
of adaptive responses to future climate and associated MHWs.

## 6.   Conclusion

Utilising the EcoTroph-Dyn trophodynamic framework for MHWs, we highlighted substantial
and latent repercussions of MHWs, which are particularly consequential for higher TLs. As a result, the
recovery/restoration time can extend over several years, if not decades. However, considering the
dynamics and characteristics of current and future MHWs, it can be anticipated that ecosystems might
not be afforded the necessary temporal window to recover between successive MHW events, which
can significantly disrupt long-term trends associated with climate change.

## 7.   Code availability

The code for the EcoTroph_Dyn model that supports the findings of this study is openly
available at https://doi.org/10.57745/NHVPCR .

## 8.   Data availability

Daily SST observations from the NOAA _ AVHRR data are publicly available on the link
https://www.ncei.noaa.gov/access/metadata/landing-page/bin/iso?id=gov.noaa.ncdc:C00680.
Species occurrence data and associated trophic levels that support the findings of this study are openly
available at https://doi.org/10.57745/PI0N92.

## 9.   Authors contributions

V.G.D.L, W.W.L.C, and D.G undertook conceptualisation, methodology, validation, and writing
– review & editing. W.W.L.C and D.G undertook funding acquisition and supervision. V.G.D.L handled



the data curation, formal analysis, investigation, project administration, software, visualisation, and writing – original draft.

## 10. Competing interests

The authors declare that they have no conflict of interest

## 11. Acknowledgement

We thank the collaboration between L'Institut Agro, Rennes and the University of British Columbia, Vancouver. We further thank Jerome Guitton for his technical support.

## 12. Financial support

V.G.D.L and D.G acknowledges funding support from the Region Bretagne. W.W.L.C and V.G.D.L acknowledge funding support from the NSERC Discovery Grant and the SSHRC through the Solving-FCB partnership.

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
