# Peer review of "Marine heatwaves deeply alter marine food web structure and function"

_EGUsphere, 2024_

## Referee Comment (RC2)

**Review of „Marine heatwaves deeply alter marine food web structure and function"**

**Summary**

In this research article, the authors performed global hindcast simulations with the EcoTroph-Dyn numerical model to estimate the distinct impacts of marine heat waves (MHWs) on the trophodynamics of marine ecosystems. They found that MHWs generally lead to a decrease in biomass, with the decrease being stronger and longer lasting for higher trophic levels. They conclude that in the future, ecosystems may not be able to recover between successive MHW events, which may disrupt trends associated with long-term climate change.

**General comments**

Overall, the manuscript is coherently written and provides novel insights into an important and timely topic. However, the introduction is quite short and should be expanded to provide a better overview and deeper understanding of the topic (see specific comments). Several minor points should also be added or clarified in the Material and Methods, Results, and Discussion sections, which are nevertheless well written and understandable. The conclusions are quite short but precise; however, I think it should at least be specified which repercussions of MHWs where identified in the current study.

Linguistically, the manuscript contains some minor grammatical, typographical, and formatting issues, especially in the references, that need to be addressed. I have listed the issues I found in "Technical corrections" and also made some suggestions to improve clarity and readability.

**Specific comments**

- L. 33-48: A clear, quantitative definition of heatwaves would help this paragraph, especially since you give quantitative changes in heatwave duration, frequency etc.
- L. 49-60: This paragraph should introduce more MHW-related ecosystem modeling studies to put the current study into a broader context. Specifically, it should be made clear to the reader what has been done already and what is new about the current study. I would also recommend to place this paragraph before the last paragraph of the introduction (i.e., between l. 66 and l. 67) to create a nice transition to the description of the current study.
- L. 51-52: It is not clear which method(s) Carneiro et al. used.
- L. 62-63: Could you elaborate on this further and explain the processes behind?
- L. 78: I would leave out "proceeding their occurrences", it makes the sentence difficult to understand.
- L. 89: Is there a specific reason for using TL width = 0.1?
- L. 92-93: Why is the biomass transfer in lower TLs faster?
- L. 97: biomass flows between(?) the trophic biomass spectra
- L. 118: I don't understand what you mean by "the trophic level of each TL classes j.". Do you mean the trophic level of the $j^{th}$ TL class?
- L. 119-120: Is this assumption based on observations/experiments? Give appropriate references.
- L. 125-126: "Biomass spectra in EcoTroph-Dyn are split into trophic classes with variable widths of trophic levels." – I don't really understand this sentence. Do you mean: "Biomass spectra in EcoTroph-Dyn are split into trophic classes of variable width."?

- L. 164-165: "lasting 15 days of the fortnight" − something must be wrong here.
- L. 168: I don't understand this sentence. Do you mean: "We used an alpha of 0.2 in our simulations."?
- Figure 1: How do you derive transfer efficiency, MHW mortality, and flow kinetic from satellite data?
- L. 193-194: Why did you use a single threshold and not one for each month, for example? Which impact may the use of only one threshold have on your results?
- L.215: How can a day with $Y_t < T_t + S_t$ be an MHW day? Can you explain this in more detail?
- L. 224-229: Could you further explain the EPPLEY-VGM method? Not all readers may be familiar with this method nor the VGM method, so I think especially basic information would be helpful. For example, what is the general concept of these methods and what is $P_{b\_opt}$?
- L. 230-235: This should at least be mentioned in the Discussion (somewhere in the paragraph l. 656-675).
- L. 236-237: Which biases may be introduced by this duplication? This should also be included into Sect. 5.3.
- L. 285-287: "Furthermore, more days with MHWs with lower intensity were identified for low latitude regions (23°N - 6°S) (Figure 3c) compared to MHWs identified in higher latitude regions (> 23°N and 25°S)." This sentence is not really clear. Do you mean that the intensity of MHWs was generally lower in high latitude regions?
- Figure 3: The figure caption seems to be mixed up. The description for c) seems to match panel d), while the description for d) does not match any panel. Thus, panel c) has no fitting description.
- L. 304: "effects of the short-term impacts of MHWs on the long-term changes" This part seems a bit confusing and contradictory, I would leave out the "short-term impacts".
- L. 320: Even if explained in the caption of Fig. 4, I would also define the three biomes in the text since the figure may be placed somewhere else in the typeset paper.
- L. 337: Using numbers for biomass increase for both scenarios would make it easier to compare the results, i.e., "a biomass increase was projected to occur in 24% of the global ocean"
- L. 358: Maybe it would be useful to give the number for global biomass loss without MHWs again for direct comparison.
- L. 365: What do you mean with "expected"? You already analyzed the differential impact of MHWs on trophic levels in Sect. 4.2.2, didn't you?
- L. 377: Can you explain what an ANOVA is?
- L. 383: greatest instead of greater?
- L. 403: Why do you use different reference periods to define temperature anomalies? In this way, the anomalies are not consistent. I would suggest to choose one of both periods. Did you use the same reference periods to calculate the temperature anomalies shown in Fig. 8? If yes, these should be corrected as well.
- L. 408: The biomass decreased but the biomass loss increased
- L. 430-431: This part is difficult to understand. Do you mean: "Considering the influence of MHWs from 2013 to 2016 using the 'with MHWs' scenario and alpha=0.2 resistance capacity"?
- l. 456-459: This sentence is difficult to understand. Maybe replace with something like: "In this study, we accounted for MHWs in the last four decades using hindcast simulations and showed the potential of synergic impacts of MHWs (pulses) and long-term climate change (presses) on biomass and trophodynamics of ecosystems.".

- l. 494-495: This explanation of TE would be helpful in the methods section.
- L. 521: metabolic efficiency or transfer efficiency? Shouldn't the ratio between ingested and stored energy be high, i.e., only a small part of the ingested energy is stored and the rest is lost?
- L.529-530: Why is the mortality higher in low TLs?
- L. 534: If the increase is 1% I wouldn't use the word "sharp".
- L. 561-562: I don't understand part (ii), could you explain this further?
- L. 585-587: The structure of this sentence seems odd and makes it difficult to understand. Please check and revise.
- L. 599-600: What defines the models in this family? What do they have in common?
- L. 607-608: What are biomass density values?
- L. 615: The word "projection" usually refers to simulations/estimates for the future. Since you performed hindcast simulations, I would use a different word here to avoid confusion. Please also check the rest of the manuscript.
- L. 629-632: Can you quantitatively compare your results to those of Arimitsu et al. (2021)?
- L. 640: The reference Cheung et al. (2020) does not exist in your reference list. Do you mean Cheung & Frölicher (2020)?
- L. 647-448: This part is difficult to understand. Maybe replace with "It would therefore have been valuable to test EcoTroph-Dyn against other MHWs in the world ocean".
- L. 681: What do you mean by "dismiss any possibility"?
- L. 684-688: This sentence is quite complex and difficult to understand. Maybe replace with something like: "Even though the global impact of MHWs is negative, species-explicit modelling could improve our understanding of how various impacts of climate change and species-level responses will affect trophodynamics and ecosystem structure and function.".
- Sect. 5.4: You could highlight here how future work can build on your study in particular. For example, what analyses should your model be used for in the future? Should your model be modified/extended, and if so, how?

**Technical corrections**

- Throughout the manuscript, there are some issues with reference formatting (i.e., the use of parentheses, commas, and semicolons). I have already included a few examples below.
- L. 11:  have become
- L. 20: (NPP) data
- L. 21: observations
- L. 22: by trophic level
- L. 25:  MHW-induced decline in biomass of 8.7% ± 1.0 (standard error)  from 2013 to 2016.
- L. 27: than in lower
- L. 36: resulting in more than a doubling of the number of MHW days
- L. 37: a space is missing before the reference
- L. 43: have caused a decrease
- L. 50: numerical modelling
- L. 51: Don't use a comma for in-text citations: Carneiro et al. (2020)
- L. 66: function  have not yet been clearly understood on a global scale
- L. 68: climate change
- L. 69 and l. 70: Since this is an article with multiple authors, I would use "we" consistently.

- L. 70: MHWs (see Guibourd de Luzinais et al., 2024)
- L. 77: occurred in  the
- L. 79: Material and methods
- L. 83:  by primary producers
- L. 85: food webs
- L. 87: TLs, i.e.,
- L. 88: trophic  spectrum
- L. 90: the whole consumers biomass
- L. 92: generally being faster
- L. 94-95: the references should be put into parentheses
- L. 96: ecosystems biomass
- L. 102: MHW occurrence. EcoTroph-Dyn's algorithms details
- L 112: TL year-1
- L. 114: trophic level using
- L. 131: within the TL class [τ,τ+Δτ, is expressed as
- L. 144:  dependent
- L. 147: flow kinetic (K)) and where
- L. 149: 3.2 MHW loss rate algorithm
- L. 152: loss rate algorithm
- L. 153: into EcoTroph-Dyn
- L. 154:  (period 1982-2011)
- L. 156: Matching historical MHW distributions and characteristics with species distribution
- L. 158: Based on this percentage estimation of an additional loss rate
- L. 159-160: Finally,  we assumed in the mathematical expression of loss rate ηi that species are continuously challenged by  increased MHW intensity, which is expressed as:
- L. 164-165: with β ranging from β=0 (no MHW) to β=1 (MHW lasting 15 days of the fortnight)
- L. 165: MHWcat,i corresponds to an MHW intensity index
- L. 169: to community resistance capacity  by testing
- L. 184: without MHWs
- L. 189: every MHW day
- L. 207: seasonal component (St) is then
- L. 208: estimation of  on the trend-adjusted series
- L. 212-213: without MHWs
- L. 215: component, i.e.,
- L. 215: For MHW days with Yt below  or non-MHW-days, we keep
- L. 217: referred to here as
- L. 218:  adapt
- L. 222: when an MHW lasts for an entire fortnight.
- L. 225: is a hybrid model
- L. 239-240:  for the scenarios
- L. 246: of  12 years

- L. 254: past MHW events
- L. 267:  in the period 2015-2021
- L. 268: NPP changes  was observed
- L. 268:  In particular
- L. 270, 272: in the period 2015-2021
- L. 273:  warming by 1°C  over
- L. 277:  and the average of
- L. 292: Evolution of the spatial extent
- L. 292-293: Evolution of MHW average duration categorised by  intensity
- L. 299: on average by 0.07 ± 0.02%
- L. 317: S2 for biome spatial definition
- L. 321: with MHWs' the declines
- L. 332, L. 339, L. 358, L. 375:  in the period 2015-2021
- L. 335 and Fig. 5 caption: For the trophic level classes, the second opening parenthesis needs to be a closing parenthesis
- L. 336-337:  with the tropical and upwelling biomes being notably more impacted.
- Fig. 5: Change in trophic group biomass (y-axis)
- L. 345-346: Projected changes in consumer biomass by trophic level and biome under the 'without MHW' and 'with MHWs' scenarios relative to the 1998-2009 average .
- L. 362: off the coast of Papua New Guinea
- L. 389: by trophic level
- L. 392: 75th quantiles
- L. 395: the response of low TLs
- L. 402: in the biomass spectrum
- L. 403: relative to the 2016 average
- L. 416: exhibited a significant total consumer biomass decrease
- L. 417: the scenario with and without MHW
- L. 420: the most
- L. 420-421: Under
- L. 425: provinces were  most affected by the MHW
- L. 426: biomass decreases of 5% and 3.8%  relative to the 'without MHW' scenario
- L. 429: lower TL
- L. 433:  by 2021
- L. 443: change in the 'without MHW' and 'with MHWs' scenarios
- L. 444: indicates the duration of 'the Blob' .
- L. 451: 0.2, while
- L. 452:  in the period 2013-2016
- L. 457-458:  long-term
- L. 466, 468: Be careful with the use of past tense. The suggestions of your study have not expired, so use "suggest" instead of "suggested" in L. 466. Similar cases appear throughout the manuscript.

- L. 470: ecosystems, which is congruent with studies by Arimitsu et al. (2021), Gomes et al. (2024), and Smith et al. (2023) .
- L. 373: ecosystem perturbations
- L. 475: of the perturbation in ecosystem functioning
- L. 476-477: intensity and duration of MHWs  have continuously increased
- L. 478: hindcast period
- L. 482: recover to pre-perturbed
- L. 484: upwelling biomes, where the hindcast biomass of high TLs consistently
- L. 488: may be
- L. 490:  continued
- L. 514: , which is used
- L. 551:  MHWs
- L. 564:  in our simulations
- L. 566: in the California Ocean
- L. 568: with an increase
- L. 570:  differentially
- L. 575: Differences in  the intensity
- L. 577: subject
- L. 592-595: For example, communities in the Gulf of Alaska are more efficient than those in the Californ Current (du Pontavice et al., 2020), and the energy entering the food web was less disrupted than in the Californ Current, which may explain the greater impact of the MHW on the Californ Current.
- L. 598: MHW hindcast
- L. 602: by their trophic level
- L. 632-634: It is worth noting that projections obtained using a smaller (larger) α led to an underestimation (overestimation) of biomass losses and changes in biomass flow parameters relative to the estimates of Cheung & Frölicher (2020) and Gomes et al. (2024) .
- L. 640-641: Grey violin plots correspond to results from Cheung et al. (2020), while the red ones correspond to our hindcast EcoTroph-Dyn simulation with α=0.2 .
- L. 656: Furthermore, uncertainties in our results arise from  the environmental drivers of EcoTroph-Dyn.
- L. 657: EcoTroph-Dyn  was driven
- L. 660: In this study, we did not consider the 'with' and 'without' MHWs scenario for NPP.
- L. 663: may have overestimated MHW impacts
- L. 666: Why do you capitalize marine ecosystem models?
- L. 681-682: Running the aforementioned five MHW-induced loss rate  scenarios
- L. 683-684: with  an increasing biomass loss over marine ecosystems with decreasing resistance capacities  (increasing α ).
- L. 690:  In our study
- L. 693: anomalously low wind, an anomalously weak Ekman transport
- L. 694: north, and coupled with anomalously low air-sea heat exchange
- L. 696: have already contributed
- L. 700: better understand

---

## Author Comment (AC3)

Dear Editor and reviewer,

Thank you for allowing us to submit a revised version of our manuscript entitled: "Marine heatwaves deeply alter marine food web structure and function". Below, we provided a detailed response regarding the concerns of the reviewers and we listed the improvements made to the manuscript.

These changes will be included in the revised version of the manuscript.

Notably, we follow the guidance from the reviewers to expand the introduction and improve the discussion of our study's findings.

Sincerely,

Vianney GUIBOURD DE LUZINAIS on behalf of the coauthors,

NB: the text in blue indicates the proposed modifications to the manuscript.

**Reviewer 2 :**

| Comment to the authors: | Responses to comments |
|---|---|
| **Summary**
 In this research article, the authors performed global hindcast simulations with the EcoTroph-Dyn numerical model to estimate the distinct impacts of marine heat waves (MHWs) on the trophodynamics of marine ecosystems. They found that MHWs generally lead to a decrease in biomass, with the decrease being stronger and longer lasting for higher trophic levels. They conclude that in the future, ecosystems may not be able to recover between successive MHW events, which may disrupt trends associated with long-term climate change. | |
| **General comments**
 Overall, the manuscript is coherently written and provides novel insights into an important and timely topic. However, the introduction is quite short and should be expanded to provide a better overview and deeper understanding of the topic (see specific comments). Several minor points should also be added or clarified in the Material and Methods, Results, and Discussion sections, which are nevertheless well written and understandable. The conclusions are quite short but precise; however, I think it should at | Dear RC2, thank you for your positive general comment on the manuscript: Through the revised manuscript, grammatical errors will be amended. We agree with your opinion about the introduction, and we will expand some of the sections as proposed. Furthermore, we will clarify the minor points in the Material and Methods, Results, and Discussion sections. |

| | |
|---|---|
| least be specified which repercussions of MHWs were identified in the current study. | |
| Linguistically, the manuscript contains some minor grammatical, typographical, and formatting issues, especially in the references, that need to be addressed. I have listed the issues I found in "Technical corrections" and also made some suggestions to improve clarity and readability. | Thank you very much for pointing out these issues; we will address them in the revised manuscript. |
| **Specific comments** | |
| **Comment1:** L. 33-48: A clear, quantitative definition of heatwaves would help this paragraph, especially since you give quantitative changes in heatwave duration, frequency etc. | Thank you for the suggestion. We will specify the definition of MHW as "Over the last century, marine heatwaves (MHWs) - defined as more than 5 days period of anomalously warm sea surface temperatures (SSTs) exceeding a specific threshold, typically determined by natural climatological variations - have increased in frequency, duration, and intensity (Frölicher et al., 2018; Hobday et al., 2016)." |
| **Comment2:** L. 49-60: This paragraph should introduce more MHW-related ecosystem modeling studies to put the current study into a broader context. Specifically, it should be made clear to the reader what has been done already and what is new about the current study. I would also recommend to place this paragraph before the last paragraph of the introduction (i.e., between l. 66 and l. 67) to create a nice transition to the description of the current study. | We agree, moving this paragraph before the last paragraph of the introduction is better for transition. We will add more MHW-related ecosystem modelling studies to put the current study into a broader context; we propose from line 54 "Previous studies assessed MHWs impacts through numerical modelling approaches. For example, Cheung et al., (2021) and Cheung & Frölicher, (2020) employed climate-fish-fisheries models to investigate MHW implications for biomass and potential catches of exploited marine species and their implication for fisheries. They found that MHWs may cause biomass decreases and shifts in the biogeography of fish stocks that are faster and bigger in magnitude than the effects of decadal-scale mean changes. They projected a doubling of impact levels by 2050 amongst the most important fisheries species over previous assessments that focus only on long-term climate change. Gomes et al. (2024) use the Ecopath with Ecosim (EwE) modelling approach to assess the ecological impacts of 'the Blob' MHW. They highlighted the alteration of trophic interactions and energy flux following the MHWs which might have profound consequences for the specific ecosystem structure and function. However, there is a gap in applying ecosystem modelling framework to study global impacts of MHWs on ecosystem structure and functions. |

| | |
|---|---|
| **Comment3:** L. 51-52: It is not clear which method(s) Carneiro et al. used. | We propose the following clarification, "For example, Carneiro et al., (2020) assessed the evolution of physiological and biochemical parameters and survival rates of the clam *Anomalocardia flexuosa* in response to simulated MHWs. Under laboratory conditions, *Anomalocardia flexuosa* was allowed to adapt to a stable control condition before being exposed to simulated conditions of MHWs occurrence lasting up to 21 days by warming the tank water by 3°C above the control temperature." |
| **Comment4:** L. 62-63: Could you elaborate on this further and explain the processes behind? | We will elaborate on this part: "Trophic dynamics of marine ecosystems are affected by ocean temperature (Eddy et al., 2021; du Pontavice et al., 2019). In particular, ocean warming is expected to increase the speed of energy transfer through the food web, i.e., flow kinetic (du Pontavice et al., 2020). Faster flow kinetic under ocean warming represents the increasing dominance of short-lived species so that each unit of biomass spends less time at a given TL and, on average, across all TLs, ultimately leading to a decrease in total biomass (Gascuel et al., 2008). Simultaneously, ocean warming is expected to induce a decrease in biomass transfer efficiency, altering both consumer production and biomass due to larger energy losses between each TL (du Pontavice et al., 2019). Therefore, ocean warming alters both the amount and speed of matter and energy transfer within the food web, potentially leading to a decline in consumer biomass through independent and cumulative effects (du Pontavice et al., 2021; Guibourd de Luzinais et al., 2023)." |
| **Comment5:** L. 78: I would leave out "proceeding their occurrences", it makes the sentence difficult to understand. | Yes, we agree, we will remove this part of the sentence. |
| **Comment6:** L. 89: Is there a specific reason for using TL width = 0.1? | Previous studies applying EcoTroph employed a TL width of 0.1, primarily because of computational efficiency while maintaining a good representation of food web structure and functions. We will mention this in the revised manuscript. |
| **Comment7:** L. 92-93: Why is the biomass transfer in lower TLs faster? | It is due to the shorter life expectancy from low TL individuals. We propose this modification: "The time needed for the biomass to flow from one to the next trophic class varies along the food chain, with biomass transfers generally faster in lower TLs (as species generally have short life-expectancy) than in the higher ones.". |

| | |
|---|---|
| **Comment8:** L. 97: biomass flows between(?) the trophic biomass spectra | No, the correct term is "in" and not "between" as with EcoTroph-Dyn, we analyse biomass flow changes in each trophic biomass spectrum. By "each trophic biomass spectrum," we refer to the trophic biomass spectrum represented in each 1° × 1° longitude-latitude ocean cell. |
| **Comment9:** L. 118: I don't understand what you mean by "the trophic level of each TL classes j.". Do you mean the trophic level of the j TL class? | Yes, we meant that, we will modify the sentence in the revised manuscript. |
| **Comment10:** L. 119-120: Is this assumption based on observations/experiments? Give appropriate references. | We propose the following sentence modification: "MHWs cause marine organism mortality, impacting their life expectancy (Smith et al., 2023). In EcoTroph-Dyn, these changes in life expectancy are reflected in the loss rate within the biomass spectrum, representing the proportion of biomass that neither persists nor progresses through the food web (du Pontavice et al., 2021; Gascuel et al., 2008, Guibourd de Luzinais et al., 2024). Therefore, according to equations used in the steady-state version of EcoTroph, the flow kinetics during MHWs is increased as follows:" |
| **Comment11:** L. 125-126: "Biomass spectra in EcoTroph-Dyn are split into trophic classes with variable widths of trophic levels." – I don't really understand this sentence. Do you mean: "Biomass spectra in EcoTroph-Dyn are split into trophic classes of variable width."? | Yes, we meant that. We will modify the sentence in the revised manuscript. |
| **Comment12:** L. 164-165: "lasting 15 days of the fortnight" – something must be wrong here. | Sorry it's a misuse of language. We ran the EcoTroph-Dyn model at a half month time step, and the expression seems to be a fortnight? We define a fortnight for our study as 1/24 of a year in line 219. To avoid confusion, we propose to change the term "fortnight" to "15 days" on lines 98, 165, 173, 218, 220, 221, 222, and 237. |
| **Comment13:** L. 168: I don't understand this sentence. Do you mean: "We used an alpha of 0.2 in our simulations."? | Yes, we meant that, we will modify the sentence in the revised manuscript. |
| **Comment14:** Figure 1: How do you derive transfer efficiency, MHW mortality, and flow kinetic from satellite data? | To derive transfer efficiency, MHW mortality, and flow kinetic from satellite data, we use empirical equations where satellite data (SST) intervenes |
| **Comment15:** L. 193-194: Why did you use a single threshold and not one for each month, for example? Which impact may the use of only one threshold have on your results? | The objective of the study was to assess the impact of MHWs, to do so we preferably detect the ones where species are undergoing a thermal stress which occurs mostly during summer period. |

| | We will clarify the framework of defining MHWs in this study in the introduction: line 67 "This study aimed to disentangle the additional or synergistic consequences of MHWs occurring during the year's warmest month, when species are undergoing a thermal stress, with the effects of the slow-onset climate changes in marine ecosystems." |
|---|---|
| | Furthermore, we will incorporate the following paragraph after line 655 in the discussion. |
| | "Here, we focused solely on the direct impacts of MHWs occurring during the year's warmest month via thermal stress, resulting in species mortality (Oliver et al., 2021; Smith et al., 2023). However, MHWs in other seasons can also have consequences on populations by affecting a specific stage of the life cycle of certain species (Crickenberger & Wethey, 2018; Oliver et al., 2021; Smith & Thatje, 2013). For example, MHWs that stress adult breeders can lead to a decrease in reproductive investment and, consequently, fewer, smaller, and lower quality gametes (e.g., Shanks et al., 2020), resulting in a loss of abundance and biomass of some species (Johansen et al., 2021). While taking seasonality into account will increase the number of detected extreme events, some may not have ecological consequences (Oliver et al., 2021; Smith et al., 2023). Thus, our approach can be seen as conservative and may underestimate the impact of MHWs. Nevertheless, we detected MHWs with potentially significant ecological impacts. Studying MHWs occurring in seasons other than summer would involve considering the phenological effects of MHWs. However, since the EcoTroph-Dyn model does not directly represent this phenological aspect of marine organisms, future studies can apply other approaches that explicitly represent seasonal processes such as spawning and migration to elucidate the effects of phenology" |
| **Comment16:** L.215: How can a day with $Y_t < T_t + S_t$ be an MHW day? Can you explain this in more detail? | In our study, we can have a MHW day with $Y_t < T_t + S_t$. This comes from the use of an annual threshold value to detect MHW events (see appendix figure S5 just before April month). We propose to add line 216: "In our study, we can have a MHW day declared even though $Y_t < T_t + S_t$. This specific situation is rare (less than 0.5% of time series) and occurs because of the use of an annual threshold value to detect MHW events that mainly occurred during the year's warmest months. (See appendix figure S5 just before April |

| | month for schematic visualisation. The time series created using this algorithm is referred here as 'without MHW'." |
|---|---|
| **Comment17:** L. 224-229: Could you further explain the EPPLEY-VGM method? Not all readers may be familiar with this method nor the VGM method, so I think especially basic information would be helpful. For example, what is the general concept of these methods and what is Pb_opt? | Yes, we will add these details. We propose to develop the idea as "The EPPLEY-VGPM method is a hybrid model that employs the basic model structure and parameterisation of the standard VGPM (Vertically Generalized Production Model) computation. This model estimates net primary production (NPP) based on chlorophyll concentration, incorporating the vertical distribution of primary production. The specificity of the EPPLEY-VGPM method is that the polynomial description of the maximum daily net primary production found within a given water column (Pb_opt, expressed in units of mg carbon fixed per mg chlorophyll per hour) is replaced by the exponential relationship described by Morel (1991), based on the curvature of the temperature-dependent growth function described by Eppley (1972)." |
| **Comment18:** L. 230-235: This should at least be mentioned in the Discussion (somewhere in the paragraph l. 656-675). | We propose to add this sentence in line 673: "Secondly, in this study, in order to propose a suitable representation of the world ocean, we use an interpolation method to reconstruct an incomplete NPP time series. The interpolation was constrained by the minimum and maximum satellite data values of the NPP observed over their respective time series to ensure reliable interpolation and reduce potential bias." |
| **Comment19:** L. 236-237: Which biases may be introduced by this duplication? This should also be included into Sect. 5.3. | We propose to add this sentence after preview comment modification: "Thirdly and lastly, in this study, we duplicated NPP monthly value to able the EcoTroph-Dyn to run at a 15 days basis. This duplication may have smoothed marine ecosystem response to the historical changes in marine environment; However, it has not changed trends and conclusions of our results." |
| **Comment20:** L. 285-287: "Furthermore, more days with MHWs with lower intensity were identified for low latitude regions (23°N - 6°S) (Figure 3c) compared to MHWs identified in higher latitude regions (> 23°N and 25°S)." This sentence is not really clear. Do you mean that the intensity of MHWs was generally lower in high latitude regions? | Yes, see response to comment4 of reviewer RC1. |
| **Comment21:** Figure 3: The figure caption seems to be mixed up. The description for c) seems to match panel d), while the description for d) does | Please, see response to comment4 of reviewer RC1. |

| | |
|---|---|
| not match any panel. Thus, panel c) has no fitting description. | |
| **Comment22:** L. 304: "effects of the short-term impacts of MHWs on the long-term changes" This part seems a bit confusing and contradictory, I would leave out the "short-term impacts". | Yes, we agree, we will remove this part of the sentence. |
| **Comment23:** L. 320: Even if explained in the caption of Fig. 4, I would also define the three biomes in the text since the figure may be placed somewhere else in the typeset paper. | Ok, we will add the biomes definition in the main text, we propose to modify the text line 320 as "While total consumer biomass was projected to decrease slightly across the three biomes (temperate, tropical, and upwelling biomes represented in Figures 4b, c, and d, respectively)". |
| **Comment24:** L. 337: Using numbers for biomass increase for both scenarios would make it easier to compare the results, i.e., "a biomass increase was projected to occur in 24% of the global ocean" | We agree, we will modify the sentence in the revised manuscript. |
| **Comment25:** L. 358: Maybe it would be useful to give the number for global biomass loss without MHWs again for direct comparison. | We agree, we will add the percentage in the revised manuscript "Under the 'with MHWs' scenario, this proportion decreased and a biomass decrease was projected to occur in 76% of the global ocean, with a projected global biomass decrease of 4.8% by 2015-2021 relative to 1998-2009 compared to an only 2.4% biomass loss under 'without MHW' scenario." |
| **Comment26:** L. 365: What do you mean with "expected"? You already analyzed the differential impact of MHWs on trophic levels in Sect. 4.2.2, didn't you? | Yes, we propose to change the wording "were expected" by "are projected". |
| **Comment27:** L. 377: Can you explain what an ANOVA is? | Sure, we proposed to add an explanation in the Material & Methods section line 173: "We performed a three-way ANOVA, that is a statistical test we used to analyse the effects of trophic levels, biomes, and α value on biomass change." |
| **Comment28:** L. 383: greatest instead of greater? | No, the correct word is greater. |
| **Comment29:** L. 403: Why do you use different reference periods to define temperature anomalies? In this way, the anomalies are not consistent. I would suggest to choose one of both periods. Did you use the same reference periods to calculate the temperature anomalies shown in Fig. 8? If yes, these should be corrected as well. | We always used the reference period 1982-2011 to define temperature anomalies. We propose to rephrase the sentence 403 as "Temperature anomalies were on average >=4 °C (between 2013 to 2016) and up to 8 °C in 2015 relative to 1982-2011." |
| **Comment30:** L. 430-431: This part is difficult to understand. Do you mean: "Considering the influence of MHWs from 2013 to 2016 using the | We agree, we will modify the sentence in the revised manuscript. |

| | |
|---|---|
| 'with MHWs' scenario and alpha=0.2 resistance capacity"? | |
| **Comment31:** L. 456-459: This sentence is difficult to understand. Maybe replace with something like: "In this study, we accounted for MHWs in the last four decades using hindcast simulations and showed the potential of synergic impacts of MHWs (pulses) and long-term climate change (presses) on bio-mass and trophodynamics of ecosystems.". | We agree, we will modify the sentence in the revised manuscript. |
| **Comment32:** l. 494-495: This explanation of TE would be helpful in the methods section. | We agree, and will move it to the method section. |
| **Comment33:** L. 521: metabolic efficiency or transfer efficiency? Shouldn't the ratio between ingested and stored energy be high, i.e., only a small part of the ingested energy is stored and the rest is lost? | Thank you for pointing out this mistake, we propose to modify the sentence as "Tropical ecosystems are composed of species with low transfer efficiency (TE) (low ratio between stored energy and ingested energy), with significant energy losses that increase with temperature (Brown et al., 2004; du Pontavice et al., 2020; Schramski et al., 2015). |
| **Comment34:** L.529-530: Why is the mortality higher in low TLs? | We propose to add "as low TLs tend to have lower thermal limit than high TLs" to enhance clarity. |
| **Comment35:** L. 534: If the increase is 1% I wouldn't use the word "sharp". | We agree and will remove the word sharp in the revised manuscript. |
| **Comment36:** L. 561-562: I don't understand part (ii), could you explain this further? | We propose this modification: "(ii) the specific functioning of these ecosystems with cool water rising from depth to the surface, tends to reduce the number of MHW days compared to their adjacent open ocean (Varela et al., 2021). More generally, it has been highlighted that ocean warming does not affect coastal regions with upwelling in the same way as the open ocean (Varela et al., 2021)." |
| **Comment37:** L. 585-587: The structure of this sentence seems odd and makes it difficult to understand. Please check and revise. | Thank you, we propose the following correction: "In contrast, in the California Current biogeochemical region, the MHW was associated with a substantial increase in the abundance of pyrosomes which implied a limitation of energy flow moving toward higher trophic levels has been observed (Gomes et al., 2024)" |
| **Comment38:** L. 599-600: What defines the models in this family? What do they have in common? | We propose to rephrase the sentence to enhance clarity as "Despite its apparent simplicity and the reduced number of parameters, EcoTroph-Dyn is part of the family of "complete ecosystem models and dynamic system models" (Plagányi, 2007) as it represents all trophic levels, from primary producers to top predators." |

| | |
|---|---|
| **Comment39:** L. 607-608: What are biomass density values? | Mistake in the wording, it is simply biomass: we will delete density. |
| **Comment40:** L. 615: The word "projection" usually refers to simulations/estimates for the future. Since you performed hindcast simulations, I would use a different word here to avoid confusion. Please also check the rest of the manuscript. | We will change this to "historical simulation". |
| **Comment41:** L. 629-632: Can you quantitatively compare your results to those of Arimitsu et al. (2021)? | We can only compare qualitatively our results with those of Arimitsu et al., 2021, as in their study they work at the species level and did not account for species of the entire food-web/model, while in our study we use a trophic level-based approach. |
| **Comment42:** L. 640: The reference Cheung et al. (2020) does not exist in your reference list. Do you mean Cheung & Frölicher (2020)? | Yes, thank you for pointing out this mistake. |
| **Comment43:** L. 647-448: This part is difficult to understand. Maybe replace with "It would therefore have been valuable to test EcoTroph-Dyn against other MHWs in the world ocean". | Yes, your proposition is simpler, we will change it. |
| **Comment44:** L. 681: What do you mean by "dismiss any possibility"? | The sentence should be rewritten as "To be cautious, we considered various loss rate scenarios to obtain a complete range of responses from marine ecosystems." |
| **Comment45:** L. 684-688: This sentence is quite complex and difficult to understand. Maybe replace with something like: "Even though the global impact of MHWs is negative, species-explicit modelling could improve our understanding of how various impacts of climate change and species-level responses will affect trophodynamics and ecosystem structure and function.". | We agree and will make changes to the revised version of the manuscript. |
| **Comment46:** Sect. 5.4: You could highlight here how future work can build on your study in particular. For example, what analyses should your model be used for in the future? Should your model be modified/extended, and if so, how? | Yes EcoTroph-Dyn model can be used to project the impacts of hypothetical future MHWs at global and local scales. See response to reviewer RC1 comment5. |
| **Technical corrections** | |
| Throughout the manuscript, there are some issues with reference formatting (i.e., the use of parentheses, commas, and semicolons). I have already included a few examples below.

● L. 11: are becoming have become

● L. 20: (NPP) data | Thank you for pointing out these technical corrections line by line. We will address them through the revised version of the manuscript and make a full check of the manuscript for others. |

- L. 21: observations

- L. 22: by trophic levels

- L. 25: specific MHW-induced decline in biomass of 8.7% ± 1.0 (standard error) in the region from 2013 to 2016.

- L. 27: than in lower

- L. 36: resulting in more than a doubling of the number of MHWs days

- L. 37: a space is missing before the reference

- L. 43: have caused a decrease

- L. 50: simulationnumerical modelling

- L. 51: Don't use a comma for in-text citations: Carneiro et al., (2020)

- L. 66: function globally have not yet been clearly understood on a global scale

- L. 68: climate changes

- L. 69 and l. 70: Since this is an article with multiple authors, I would use "we" consistently.

- L. 70: MHWs (see (Guibourd de Luzinais et al., 2024)

- L. 77: occurred in on the

- L. 79: Material and methods

- L. 83: from by primary producers

- L. 85: food webs

- L. 87: TLs, i.e.,

- L. 88: trophic  spectrum

- L. 90: the whole consumers biomass

- L. 92: generally being faster

- L. 94-95: the references should be put into parentheses

- L. 96: ecosystems biomass

- L. 102: MHWs occurrence. EcoTroph-Dyn's algorithms' details

- L 112: TL. year-1

- L. 114: trophic level, using

- L. 131: within the TL class $[\tau,\tau+\Delta\tau[$. It], is expressed as

- L. 144:  dependent

- L. 147: flow kinetic (K)) and where

- L. 149: 3.2 MHW loss rate algorithm computation

- L. 152: loss rate algorithm computation

- L. 153: into EcoTroph-Dyn

- L. 154: period (period 1982-2011)

- L. 156: Matching historical MHWs' historical distributions and characteristics with species distribution

- L. 158: Estimation bBased on this percentage estimation of an additional loss rate

- L. 159-160: Finally, through loss rate (ηi) mathematical expression, we assumed in

the mathe-matical expression of loss rate $\eta i$ that species arewere continuously challenged by MHW in-creased MHW intensity, which is expressed as:

- L. 164-165: with β rangesranging from β=0; (no MHW), to β=1; (MHW lasting 15 days of the fortnight)
- L. 165: MHWcat,i corresponds to an MHW intensity index
- L. 169: to community resistance capacity to MHW by testing
- L. 184: without MHWs
- L. 189: every MHWs day
- L. 207: seasonal component ($St$), is then
- L. 208: estimation of ($St$) on the trend-adjusted series
- L. 212-213: without MHWs
- L. 215: component, i.e.,
- L. 215: For MHW days with $Yt$ below ($Tt + St$) or not an MHW daynon-MHW-days, we keep
- L. 217: referred to here as
- L. 218: match adapt
- L. 222: when an MHW lasts for an entire fortnight.
- L. 225: is a hybrid model
- L. 239-240: under for the scenarios

- L. 246: of simulating 12 years

- L. 254: past MHWs events

- L. 267: by in the period 2015-2021

- L. 268: NPP changes were was observed

- L. 268: Notably In particular

- L. 270, 272: in the period 2015-2021

- L. 273: warmed up warming by 1°C during over

- L. 277: relative to the average between and the average of

- L. 292: Evolution of the spatial extent

- L. 292-293: Evolution of MHWs averaged duration categorised by their intensity

- L. 299: on average, by $0.07 \pm 0.02\%$

- L. 317: S2 for biomes spatial definition

- L. 321: with MHWs', the declines

- L. 332, L. 339, L. 358, L. 375: by in the period 2015-2021

- L. 335 and Fig. 5 caption: For the trophic level classes, the second opening parenthesis needs to be a losing parenthesis

- L. 336-337: notably with the tropical and upwelling biomes being notably more impacted.

- Fig. 5: Change in trophic groups biomass (y-axis)
- L. 345-346: Projected changes in consumer biomass by trophic levels and biomes under the 'without MHW' and 'with MHWs' scenarios relative to the 1998-2009 average between 1998-2009.
- L. 362: off the coast of Papua New Guinea Coast
- L. 389: by trophic levels
- L. 392: 75th quantiles
- L. 395: the response of low TLs response
- L. 402: in the biomass spectrum
- L. 403: relative to the 2016 average
- L. 416: exhibited a significant total consumer biomass decrease
- L. 417: the scenarios with and without MHWs
- L. 420: the most
- L. 420-421: However, uUnder
- L. 425: provinces were the most affected by the MHW
- L. 426: biomass decreases of 5% and 3.8% 'with MHWs' relative to the 'without MHW' scenario
- L. 429: lower TLs
- L. 433: as of by 2021

- L. 443: change in the 'without MHW' and 'with MHWs' scenarios

- L. 444: indicates the duration of 'the Blob' duration.

- L. 451: 0.2, while

- L. 452: by in the period 2013-2016

- L. 457-458: longterm long-term

- L. 466, 468: Be careful with the use of past tense. The suggestions of your study have not expired, so use "suggest" instead of "suggested" in L. 466. Similar cases appear throuhout the manuscript

- L. 470: ecosystems, which is congruent with studies by Arimitsu et al., (2021);, Gomes et al., (2024);, and Smith et al., (2023) studies.

- L. 373: ecosystems perturbations

- L. 475: of the perturbation in ecosystem functioning perturbation

- L. 476-477: intensity and duration of MHWs characteristics have continuously increased

- L. 478: hindcast period hindcast

- L. 482: recover to pre-perturbed

- L. 484: upwelling biomes, where the hindcast biomass of high TLs consistently

- L. 488: may, therefore, be

- L. 490: continuing continued

- L. 514: , which is used

- L. 551: These MHWs

- L. 564: through our modelling approach in our simulations

- L. 566: in the California Ocean

- L. 568: with anthe increase

- L. 570: differently differentially

- L. 575: Differences in exposure to the intensity

- L. 577: subjected

- L. 592-595: For example, communities in the Gulf of Alaskan Gulf are more efficient than those in the Californian Current (du Pontavice et al., 2020), and the energy entering the food web was less disrupted than in the Californian Current, which may explain the greater impact of the MHW on the Californian Current.

- L. 598: MHWs hindcast

- L. 602: by their trophic levels

- L. 632-634: It is worth noting that projections obtained fromusing a smaller (larger) α led to an underestimation (overestimation) of biomass losses and changes in biomass flow parame-ters

- relative to the estimates of Cheung & Frölicher, (2020) and Gomes et al., (2024) esti-mates.

- L. 640-641: Grey violin plots correspond to results from Cheung et al., (2020), while the red ones corresponds to our hindcasted EcoTroph-Dyn simulation with α=0.2 scenario.

- L. 656: Furthermore, uncertainties aboutin our results arise from EcoTroph-Dyn the environmental drivers of EcoTroph-Dyn.

- L. 657: EcoTroph-Dyn has been was driven

- L. 660: In this study, we did not consider the 'with' and 'without' MHWs scenarios for NPP.

- L. 663: couldmay have overestimated MHWs impacts

- L. 666: Why do you capitalize marine ecosystem models?

- L. 681-682: Running the aforementionedthese five MHW-induced loss rate-induced scenarios

- L. 683-684: with a worsening an increasing biomass loss over marine ecosystems with de-creasing resistance

capacities decreasing (increasing α increasing).

- L. 690: From In our study
- L. 693: anomalously low wind, an anomalously weak Ekman transports
- L. 694: north, and, coupled with anomalously low air-sea heat exchanges
- L. 696: have already contributed
- L. 700: better understand better

---

## Author Comment (AC4)

Dear Editor and reviewer,

Thank you for allowing us to submit a revised version of our manuscript entitled: "Marine heatwaves deeply alter marine food web structure and function". Below, we provided a detailed response regarding the concerns of the reviewers and we listed the improvements made to the manuscript.

These changes will be included in the revised version of the manuscript.

Notably, we follow the guidance from the reviewers to expand the introduction and improve the discussion of our study's findings.

Sincerely,

Vianney GUIBOURD DE LUZINAIS on behalf of the coauthors,

NB: the text in blue indicates the proposed modifications to the manuscript.

**Reviewer 1 :**

| Comment to the authors: | Responses to comments |
|---|---|
| **General comments**
This article describes the application of an ecosystem dynamics model to a global sea temperature and NPP dataset to assess the occurrence of marine heatwaves and model the impacts of these heatwaves on ecosystems.
The article is concisely written and coherent overall, though there are minor grammatical errors throughout that should be amended to improve the readability and flow of the article. There are also a couple of errors with figure captioning and formatting. Some sections within the introduction and discussion could be further expanded, as described below. | Dear RC1, thank you for your positive general comment on the manuscript. Through the revised manuscript, grammatical errors will be amended as the figures' captioning and formatting. We agree with your opinion about the introduction and discussion, and we will expand some of the sections as proposed. |
| **Specific comments** | |
| **Comment1:** The abstract is effective. | Thank you very much for your comment. |
| **Comment2:** The introduction section is quite short, and could be expanded with more examples and some deeper explanations, but provides a concise and generally effective overview of the topic. In general, it would be helpful to give more information about why MHWs have such sizable ecological consequences. For example, you could discuss why changes in temperature cause stress and how organisms respond to this (see DOI: 10.1126/science.1163156), as well as the fact | Thank you for the insights, we will integer these aspects in the introduction from line 41 in a new paragraph and propose the following: "Marine ectotherms' physiological functions are directly affected by ocean temperature changes that are closely related to their body temperature (Pörtner et Farrell, 2008, Guibourd de Luzinais et al., 2024). These species are adapted to perform optimally at a range of body temperature, with certain upper and lower temperature limits within which they can survive (Pörtner et Farrell, |

| | |
|---|---|
| that temperatures may be more likely to exceed critical thresholds during MHWs (see DOI: 10.1016/j.tree.2021.09.003). | 2008). When environmental temperatures exceed this temperature range, e.g., during MHWs, the organism is stressed, leading to functional constraints and declines in performance (Pörtner et Farrell, 2008). Particularly, abnormally high temperatures during MHWs often exceed organisms' thermal limits, impacting their distribution, growth and survival (Smale et al., 2019; Smith et al., 2023, Guibourd de Luzinais et al., 2024). Moreover, impacts of MHWs at population level have cascading effects at community and ecosystems level. For example, MHW-induced declines in phytoplankton biomass and diversity have led to significant changes in zooplankton and other marine invertebrate diversity and biomass (Cavole et al., 2016). MHWs cause coral bleaching that also impacts coral reef ecosystems (Garrabou et al., 2009, 2022; Pearce et al., 2011). Range shifts driven by MHWs result in "tropicalization" of fish communities (Wernberg et al., 2016). Ultimately, MHWs imply mass mortality of fish and invertebrates modifying ecosystem functioning (Cannell et al., 2019; Cavole et al., 2016; Collins et al., 2019). However, these ecological impacts of MHWs are not ubiquitous and vary largely between MHW events, species and ecosystems (Fredston et al., 2023; Oliver et al., 2021; Pershing et al., 2018; Smale et al., 2019; Smith et al., 2023).". |
| **Comment3:** The Material and Method and Results sections are generally well-written. See below for grammatical corrections. | Thank you, we will check and correct grammatical errors. |
| **Comment4**: Figure 3 - the caption needs to be more specific about what each panel represents. For Figure 3a, the caption should specify how the spatial extent of MHWs was defined. Is this the percentage of the oceans' total surface area that experienced a MHW during each year? Or the average spatial extent of each individual MHW event?

The figure keys state that Figure 3c depicts the average number of MHW days in each cell and Figure 3d depicts the average intensity of MHWs, but the figure caption states the opposite. It is also unclear whether Figure 3c depicts the average duration of each individual MHW event in days, or the total number of MHW days per year in each cell. | We acknowledge the paragraph corresponding to fig 3 and fig 3 caption needs to be clarified. We propose these modifications:
For the figure: Fig 3C legend "" "Average MHWs duration in days"
Caption: "**Figure 3: Temporal and spatial characteristics of MHWs identified for the period 1998 to 2021.** (a) Changes in the percentage of the oceans' total surface area with MHW in each year categorised by their intensity, (b) Changes in MHWs averaged duration categorised by their intensity, and (c) Average duration of each MHW event in days that occurred over the period 2015-2021. (d) Average intensity of each MHW event over the period 2015-2021." |

| | |
|---|---|
| | For the paragraph "Under the 'with MHWs' scenario, MHWs occurring during the year's warmest month increased in intensity, duration, and surface extent from 1998 to 2021 (Figures 3a, b) with large spatial variability (Figures 3c, d). MHWs with intensity lower than 3°C above the climatology were identified on average in 28.5 % of the ocean surface (Figures 3a). These MHWs lasted, on average, more than 40 days (Figures 3b). In contrast, MHWs characterised as higher intensity (≥3°C above climatology) were identified in <20% of the ocean surface area (Figures 3a). These relatively more intensive MHWs lasted, on average, 32 days (Figures 3b). Furthermore, more MHW days of lower intensity were identified for low latitude regions (23°N - 6°S) (Figure 3c, 3d) compared to MHW days identified in higher latitude regions (> 23°N and 25°S). " |
| **Comment5:** The discussion is generally well-written and explains the findings and implications of this work with an appropriate level of detail. It would be interesting to include some recommendations for future development and use of the EcoTroph-Dyn model. For example, do you think the model could be used to predict the impacts of hypothetical future MHWs at global and local scales, and how much caution should be used when interpreting these findings? | Yes, EcoTroph-Dyn model can be used to project the impacts of MHWs at global and local scales under future scenarios. We propose to add this idea from line 705: "The EcoTroph Dyn model is a tool to understand the ecological consequences of MHWs at global and local scales, and to project their impacts under future scenarios. However, the model focuses on aggregated energy flows between trophic groups while ecological responses to MHWs between species within each group may vary substantially. Some species may acclimatize or adapt to MHWs. Consideration of the potential acclimatization/adaptation in the model requires the development of specific adaptation scenarios and model settings in addition to the model settings presented here.". |
| **Comment6:** Line 622 - do you have any ideas of why the model might have underestimated ecosystem responses to MHWs? Do you have any recommendations for how people using this model could account for this uncertainty? | Yes, the choice of the parameter values for α (representing marine communities' resistance capacity to MHW) strongly affect the sensitivity of the simulated ecosystem responses to MHWs. In this study, we used a range of α values (0.2, 0.5, and 1) and showed that an α value of 0.2 underestimates ecosystem response to 'the Blob" MHW, while an α value of 0.5 overestimates ecosystem response. To account for this uncertainty, we recommend that future study can calibrate α values for each ocean regions/ marine ecosystems based of |

| | historical MHWs impacts on marine ecosystems' biomass. |
|---|---|
| | We propose to add after line 622, "The underestimation of ecological responses to MHWs is likely caused by the choice of a lower α value that lowers the sensitivity of the ecosystem to MHWs. To reduce the uncertainty over the α value, future studies could calibrate it for specific region using observational data of MHWs impacts on marine ecosystems' biomass." |
| **Comment7:** Line 676 - Do you think it would be possible to design a species-specific or ecosystem-specific version of the EcoTroph-Dyn model that could more precisely predict the impacts of MHWs on specified regions or ecosystem types? | The power of the EcoTroph-Dyn model lies in its ability to represent the functioning of ecosystems in a general way at the trophic level scale. As mentioned in this paragraph, in order to obtain more accurate projections of the response of specific species or ecosystems, the use of more complex models operating at the species level and/or integrating more fully the physical changes in the environment during MHWs with the inclusion of other environmental variables such as $O^2$, salinity and pH would be necessary. |
| **Comment8:** Line 681 - Dismiss any possibility of what? | The sentence should be rewritten as "To be cautious, we considered various loss rate scenarios to obtain a complete range of responses from marine ecosystems." |
| **Comment9:** The conclusion section is very short. It might be useful to include a brief summary of your findings regarding the accuracy of the EcoTroph-Dyn when model compared to real-world data from 'the Blob'. | We agree and will include a brief summary of our findings regarding the accuracy of the EcoTroph-Dyn when the model is compared to real-world data from 'the Blob' into the conclusion. |
| | We propose this new conclusion: |
| | "Utilising the EcoTroph-Dyn trophodynamic framework for MHWs, we highlighted substantial and latent repercussions of MHWs, notably biomass loss and biomass flow alteration, which are particularly consequential for higher TLs. As a result, the recovery/restoration time can extend over several years, if not decades. EcoTroph-Dyn model demonstrates its capacity to characterize the impacts of MHWs on ecosystem structure and functions, with a slight underestimation of the magnitude of the impacts when the model is applied to examine 'the Blob' MHW. However, considering the dynamics and characteristics of current and future MHWs, it can be anticipated that ecosystems might not be afforded the necessary temporal window to recover between successive MHW events, which can significantly |

| | |
|---|---|
| | disrupt long-term trends associated with climate change." |
| **Comment10: Technical and grammatical corrections** | Thank you for pointing out these grammatical corrections line by line and issues with the figures captions/ format. We will address them through the revised version of the manuscript. |
| Line 11-12 - This sentence mixes present and past tense in a way that doesn't completely make sense; "have become longer" might sound better.
 ● Line 46 - Verb tenses are inconsistent; "are not ubiquitous and have varied largely" would sound more consistent. Additionally, it could be informative to provide more specific details about how ecological impacts have varied between different MHW events.

 ● Line 69 - This sentence begins with "I used" while the rest of this paragraph uses "We" - it would be better to change this to "We used" for consistency.

 ● Line 70 - There are two opening brackets in this sentence, but only one is needed. Also, MHWs rather than MHW.

 ● Line 90 - "the total biomass of all consumers" is clearer than "whole consumers biomass"

 ● Line 94-95 - The list of examples should be enclosed in brackets.

 ● Line 118 - "each TL class". | |

- Line 130 - "represents" rather than "representing"
- Line 222 - "MHWs last" or "MHW conditions last"
- Line 225 - "is a hybrid model"
- Line 267 - "large spatial variability was observed in NPP changes" is grammatically clearer
- Line 403 - this line should use the ≥ (greater than or equal to) symbol
- Line 426 - "the 'without MHW' scenario"
- Line 430 - "the 'with MHWs' scenario"
- Line 470 - "congruent with the findings of…" would be grammatically clearer
- Line 471 - "ecosystem functions"
- Line 473 - "ecosystem perturbations"
- Line 476 - "the intensity and duration of MHWs have continuously increased"
- Line 482-483 - "high TL biomass experienced greater impacts from MHWs, and was not able to recover to pre-perturbation levels as effectively as the low and medium TL biomass"
- Line 585-587 - This sentence is unclear - I assume that what you mean is "the MHW was associated with a substantial

| increase in the abundance of pyrosomes limiting/stopping energy flow moving toward higher trophic levels (Gomes et al., 2024).", but the grammatical structure of the sentence as written makes it somewhat difficult to follow.
● Figure 9 - the category labels on the X-axis are not vertically aligned with the violin plots. | |
|---|---|

---

## Author Response (AR1)

Dear Editor and reviewer,

Thank you for allowing us to submit a revised version of our manuscript entitled: "Marine heatwaves deeply alter marine food web structure and function". Below, we provided a detailed response regarding the concerns of the reviewers and we listed the improvements made to the manuscript.

These changes are now included in the revised version of the manuscript.

Notably, we followed the guidance from the reviewers to expand the introduction and improve the discussion of our study's findings.

Sincerely,

Vianney GUIBOURD DE LUZINAIS on behalf of the coauthors,

consequences. For example, you could discuss

why changes in temperature cause stress and

NB: the text in blue indicates the modifications made to the manuscript. I indicated the line numbers of the Revised Manuscript with the Track Changes version.

**Responses to comments**

al., 2024). These species are adapted to perform

optimally at a range of body temperature, with

**Reviewer 1:**

Comment to the authors:

**General comments** Dear RC1, thank you for your positive general This article describes the application of an comment on the manuscript. Through the ecosystem dynamics model to a global sea revised manuscript, grammatical errors have temperature and NPP dataset to assess the been amended as the figures' captioning and occurrence of marine heatwaves and model the formatting. We agree with your opinion about impacts of these heatwaves on ecosystems. the introduction and discussion, and we have The article is concisely written and coherent expanded some of the sections as proposed. overall, though there are minor grammatical errors throughout that should be amended to improve the readability and flow of the article. There are also a couple of errors with figure captioning and formatting. Some sections within the introduction and discussion could be further expanded, as described below. **Specific comments** Comment1: The abstract is effective. Thank you very much for your comment. Comment2: The introduction section is quite Thank you for the insights, we have included short, and could be expanded with more these aspects in the introduction from line 45 in examples and some deeper explanations, but a new paragraph as following: provides a concise and generally effective ectotherms' physiological functions are directly overview of the topic. In general, it would be affected by ocean temperature changes that are helpful to give more information about why closely related to their body temperature MHWs have such sizable ecological (Pörtner et Farrell, 2008, Guibourd de Luzinais et**

how organisms respond to this (see DOI: 10.1126/science.1163156), as well as the fact that temperatures may be more likely to exceed critical thresholds during MHWs (see DOI: 10.1016/j.tree.2021.09.003).

certain upper and lower temperature limits within which they can survive (Pörtner et Farrell, 2008). When environmental temperatures exceed this temperature range, e.g., during MHWs, the organism is stressed, leading to functional constraints and declines performance (Pörtner et Farrell, 2008). Particularly, abnormally high temperatures during MHWs often exceed organisms' thermal limits, impacting their distribution, growth and survival (Smale et al., 2019; Smith et al., 2023, Guibourd de Luzinais et al., 2024). Moreover, impacts of MHWs at population level have cascading effects at community and ecosystems level. For example, MHW-induced declines in phytoplankton biomass and diversity have led to significant changes in zooplankton and other marine invertebrate diversity and biomass (Cavole et al., 2016). MHWs cause coral bleaching that also impacts coral reef ecosystems (Garrabou et al., 2009, 2022; Pearce et al., 2011). Range shifts driven by MHWs result "tropicalization" of fish communities (Wernberg et al., 2016). Ultimately, MHWs imply mass mortality of fish and invertebrates modifying ecosystem functioning (Cannell et al., 2019; Cavole et al., 2016; Collins et al., 2019). However, these ecological impacts of MHWs are not ubiquitous and vary largely between MHW events, species and ecosystems (Fredston et al., 2023; Oliver et al., 2021; Pershing et al., 2018; Smale et al., 2019; Smith et al., 2023).".

**Comment3:** The Material and Method and Results sections are generally well-written. See below for grammatical corrections.

Thank you, we have checked and corrected the grammatical errors.

**Comment4:** Figure 3 - the caption needs to be more specific about what each panel represents. For Figure 3a, the caption should specify how the spatial extent of MHWs was defined. Is this the percentage of the oceans' total surface area that experienced a MHW during each year? Or the average spatial extent of each individual MHW event?

The figure keys state that Figure 3c depicts the average number of MHW days in each cell and Figure 3d depicts the average intensity of MHWs, but the figure caption states the opposite. It is also unclear whether Figure 3c depicts the average duration of each individual MHW event in days, or the total number of MHW days per year in each cell.

Thank you, we have revised the figure caption and the paragraph in the manuscript as follows: For the figure: Fig 3C legend "Average of MHW days" "Average MHWs duration in days"

Caption: "Figure 3: Temporal and spatial characteristics of MHWs identified for the period 1998 to 2021. (a) Changes in the percentage of the oceans' total surface area with MHW in each year categorised by their intensity, (b) Changes in MHWs averaged duration categorised by their intensity, and (c) Average duration of each MHW event in days that occurred over the period 2015-2021. (d) Average intensity of each MHW event over the period 2015-2021."

For the paragraph (line438-445) "Under the 'with MHWs' scenario, MHWs occurring during the year's warmest month increased in intensity, duration, and surface extent from 1998 to 2021 (Figures 3a, b) with large spatial variability (Figures 3c, d). MHWs with intensity lower than 3°C above the climatology were identified on average in 28.5 % of the ocean surface (Figures 3a). These MHWs lasted, on average, more than 40 days (Figures 3b). In contrast, MHWs characterised as higher intensity (≥3°C above climatology) were identified in <20% of the ocean surface area (Figures 3a). These relatively more intensive MHWs lasted, on average, 32 days (Figures 3b). Furthermore, more MHW days of lower intensity were identified for low latitude regions (23°N - 6°S) (Figure 3c, 3d) compared to MHW days identified in higher latitude regions (> 23°N and 25°S). In addition, the intensity of MHWs was higher in higher latitude regions in the northern hemisphere relative to those in the southern hemisphere (Figure 3d)."

Comment5: The discussion is generally well-written and explains the findings and implications of this work with an appropriate level of detail. It would be interesting to include some recommendations for future development and use of the EcoTroph-Dyn model. For example, do you think the model could be used to predict the impacts of hypothetical future MHWs at global and local scales, and how much caution should be used when interpreting these findings?

Yes, EcoTroph-Dyn model can be used to project the impacts of MHWs at global and local scales under future scenarios. We have incorporated this idea from line 810:

"The EcoTroph Dyn model is a tool to understand the ecological consequences of MHWs at global and local scales, and to project their impacts under future scenarios. However, the model focuses on aggregated energy flows between trophic groups while ecological responses to MHWs between species within each group may vary substantially. Some species may acclimatize or adapt to MHWs. Consideration of the potential acclimatization/adaptation in the model requires the development of specific adaptation scenarios and model settings in addition to the model settings presented here.".

**Comment6:** Line 622 - do you have any ideas of why the model might have underestimated ecosystem responses to MHWs? Do you have any recommendations for how people using this model could account for this uncertainty?

Yes, the choice of the parameter values for  $\alpha$  (representing marine communities' resistance capacity to MHW) strongly affect the sensitivity of the simulated ecosystem responses to MHWs. In this study, we used a range of  $\alpha$  values (0.2, 0.5, and 1) and showed that an  $\alpha$  value of 0.2 underestimates ecosystem response to 'the Blob" MHW, while an  $\alpha$  value of 0.5 overestimates ecosystem response.

To account for this uncertainty, we recommend that future study can calibrate  $\alpha$  values for each

ocean regions/ marine ecosystems based of historical MHWs impacts on marine ecosystems' biomass.

We have added these sentences (lines 698-701) to discuss these points: "The underestimation of ecological responses to MHWs is likely caused by the choice of a lower  $\alpha$  value that lowers the sensitivity of the ecosystem to MHWs. To reduce the uncertainty over the  $\alpha$  value, future studies could calibrate it for specific region using observational data of MHWs impacts on marine ecosystems' biomass."

**Comment7:** Line 676 - Do you think it would be possible to design a species-specific or ecosystem-specific version of the EcoTroph-Dyn model that could more precisely predict the impacts of MHWs on specified regions or ecosystem types?

The power of the EcoTroph-Dyn model lies in its ability to represent the functioning of ecosystems in a general way at the trophic level scale. As mentioned in this paragraph, in order to obtain more accurate projections of the response of specific species or ecosystems, the use of more complex models operating at the species level and/or integrating more fully the physical changes in the environment during MHWs with the inclusion of other environmental variables such as O², salinity and pH would be necessary.

**Comment8:** Line 681 - Dismiss any possibility of what?

We have rewritten the sentence as follows (line 781) "To be cautious, we considered various loss rate scenarios to obtain a complete range of responses from marine ecosystems." to enhance clarity

**Comment9:** The conclusion section is very short. It might be useful to include a brief summary of your findings regarding the accuracy of the EcoTroph-Dyn when model compared to real-world data from 'the Blob'.

We agree and have included a brief summary of our findings regarding the accuracy of the EcoTroph-Dyn when the model is compared to real-world data from 'the Blob' into the conclusion.

We have rewritten the conclusion as follows: "Utilising the EcoTroph-Dyn trophodynamic MHWs, we highlighted framework for substantial and latent repercussions of MHWs, notably biomass loss and biomass flow alteration, which are particularly consequential for higher TLs. As result, the recovery/restoration time can extend over several years, if not decades. EcoTroph-Dyn model demonstrates its capacity to characterize the impacts of MHWs on ecosystem structure and functions, with a slight underestimation of the magnitude of the impacts when the model is applied to examine 'the Blob' MHW. However, considering the dynamics and characteristics of current and future MHWs, it can be anticipated

|                                                                                                                                               | that ecosystems might not be afforded the necessary temporal window to recover between successive MHW events, which can significantly disrupt long-term trends associated with climate change." |
|-----------------------------------------------------------------------------------------------------------------------------------------------|-------------------------------------------------------------------------------------------------------------------------------------------------------------------------------------------------|
| Comment10: Technical and grammatical corrections                                                                                              | Thank you for pointing out these grammatical corrections line by line and issues with the figures captions/ format. We have addressed them through the revised version of the manuscript.       |
| Line 11-12 - This sentence mixes present and past tense in a way that doesn't completely make sense; "have become longer" might sound better. |                                                                                                                                                                                                 |
| • Line 46 - Verb tenses are inconsistent;                                                                                                     |                                                                                                                                                                                                 |
| "are not ubiquitous and have varied                                                                                                           |                                                                                                                                                                                                 |
| largely" would sound more consistent.                                                                                                         |                                                                                                                                                                                                 |
| Additionally, it could be informative to                                                                                                      |                                                                                                                                                                                                 |
| provide more specific details about how                                                                                                       |                                                                                                                                                                                                 |
| ecological impacts have varied between                                                                                                        |                                                                                                                                                                                                 |
| different MHW events.                                                                                                                         |                                                                                                                                                                                                 |
| • Line 69 - This sentence begins with "I                                                                                                      |                                                                                                                                                                                                 |
| used" while the rest of this paragraph                                                                                                        |                                                                                                                                                                                                 |
| uses "We" - it would be better to change                                                                                                      |                                                                                                                                                                                                 |
| this to "We used" for consistency.                                                                                                            |                                                                                                                                                                                                 |
| Line 70 - There are two opening brackets                                                                                                      |                                                                                                                                                                                                 |
| in this sentence, but only one is needed.                                                                                                     |                                                                                                                                                                                                 |
| Also, MHWs rather than MHW.                                                                                                                   |                                                                                                                                                                                                 |
| • Line 90 - "the total biomass of all                                                                                                         |                                                                                                                                                                                                 |
| consumers" is clearer than "whole                                                                                                             |                                                                                                                                                                                                 |
| consumers biomass"                                                                                                                            |                                                                                                                                                                                                 |

- Line 94-95 The list of examples should be enclosed in brackets.
- Line 118 "each TL class".
- Line 130 "represents" rather than
   "representing"
- Line 222 "MHWs last" or "MHW conditions last"
- Line 225 "is a hybrid model"
- Line 267 "large spatial variability was observed in NPP changes" is grammatically clearer
- Line 403 this line should use the ≥
   (greater than or equal to) symbol
- Line 426 "the 'without MHW' scenario"
- Line 430 "the 'with MHWs' scenario"
- Line 470 "congruent with the findings of..." would be grammatically clearer
- Line 471 "ecosystem functions"
- Line 473 "ecosystem perturbations"
- Line 476 "the intensity and duration of MHWs have continuously increased"
- Line 482-483 "high TL biomass experienced greater impacts from MHWs, and was not able to recover to pre-perturbation levels as effectively as the low and medium TL biomass"

- Line 585-587 This sentence is unclear I assume that what you mean is "the MHW was associated with a substantial increase in the abundance of pyrosomes limiting/stopping energy flow moving toward higher trophic levels (Gomes et al., 2024).", but the grammatical structure of the sentence as written makes it somewhat difficult to follow.
- Figure 9 the category labels on the Xaxis are not vertically aligned with the violin plots.

**Reviewer 2:**

| Comment to the authors:                                                                                                                                                                                                                                                                                                                                                                                                                                                                                                                                   | Responses to comments                                                                                                                                                                                                                                                                                           |
|-----------------------------------------------------------------------------------------------------------------------------------------------------------------------------------------------------------------------------------------------------------------------------------------------------------------------------------------------------------------------------------------------------------------------------------------------------------------------------------------------------------------------------------------------------------|-----------------------------------------------------------------------------------------------------------------------------------------------------------------------------------------------------------------------------------------------------------------------------------------------------------------|
| Summary In this research article, the authors performed global hindcast simulations with the EcoTroph-Dyn numerical model to estimate the distinct impacts of marine heat waves (MHWs) on the trophodynamics of marine ecosystems. They found that MHWs generally lead to a decrease in biomass, with the decrease being stronger and longer lasting for higher trophic levels. They conclude that in the future, ecosystems may not be able to recover between successive MHW events, which may disrupt trends associated with long-term climate change. |                                                                                                                                                                                                                                                                                                                 |
| General comments Overall, the manuscript is coherently written and provides novel insights into an important and timely topic. However, the introduction is quite short and should be expanded to provide a better overview and deeper understanding of the topic (see specific comments). Several minor                                                                                                                                                                                                                                                  | Dear RC2, thank you for your positive general comment on the manuscript: Through the revised manuscript, grammatical errors have been amended. We agree with your opinion about the introduction, and we have expanded some of the sections as proposed. Furthermore, we have clarified the minor points in the |

points should also be added or clarified in the Material and Methods, Results, and Discussion sections, which are nevertheless well written and understandable. The conclusions are quite short but precise; however, I think it should at least be specified which repercussions of MHWs were identified in the current study.

Material and Methods, Results, and Discussion sections.

Linguistically, the manuscript contains some minor grammatical, typographical, and formatting issues, especially in the references, that need to be addressed. I have listed the issues I found in "Technical corrections" and also made some suggestions to improve clarity and readability.

Thank you very much for pointing out these issues; we have addressed them in the revised manuscript.

**Specific comments**

**Comment1:** L. 33-48: A clear, quantitative definition of heatwaves would help this paragraph, especially since you give quantitative changes in heatwave duration, frequency etc.

Thank you for the suggestion. We have specified the definition of MHW on line 34 as follows "Over the last century, marine heatwaves (MHWs) - defined as more than 5 days period of anomalously warm sea surface temperatures (SSTs) exceeding a specific threshold, typically determined by natural climatological variations - have increased in frequency, duration, and intensity (Frölicher et al., 2018; Hobday et al., 2016)."

**Comment2:** L. 49-60: This paragraph should introduce more MHW-related ecosystem modeling studies to put the current study into a broader context. Specifically, it should be made clear to the reader what has been done already and what is new about the current study. I would also recommend to place this paragraph before the last paragraph of the introduction (i.e., between I. 66 and I. 67) to create a nice transition to the description of the current study.

We agree, moving this paragraph before the last paragraph of the introduction is better for transition. We also have added, from line 71, MHW-related ecosystem modelling studies to put the current study into a broader context. "Previous studies assessed MHWs through numerical impacts modelling approaches. For example, Cheung et al., (2021) and Cheung & Frölicher, (2020) employed climate-fish-fisheries models to investigate MHW implications for biomass and potential catches of exploited marine species and their implication for fisheries. They found that MHWs may cause biomass decreases and shifts in the biogeography of fish stocks that are faster and bigger in magnitude than the effects of decadalscale mean changes. They projected a doubling of impact levels by 2050 amongst the most important fisheries species over previous assessments that focus only on long-term climate change. Gomes et al. (2024) use the Ecopath with Ecosim (EwE) modelling approach to assess the ecological impacts of 'the Blob' MHW. They highlighted the alteration of trophic interactions and energy flux following the MHWs

|                                                 | which might have profound consequences for
the specific ecosystem structure and function. |
|-------------------------------------------------|----------------------------------------------------------------------------------------------|
|                                                 | However, there is a gap in applying ecosystem                                                |
|                                                 | modelling framework to study global impacts of                                               |
|                                                 | MHWs on ecosystem structure and functions.                                                   |
| Comment3: L. 51-52: It is not clear which       | We have rewritten this part (line 67) to enhance                                             |
| method(s) Carneiro et al. used.                 | clarity as follows: "For example, Carneiro et al.,                                           |
| method(s) cameno et al. asea.                   | (2020) assessed the evolution of physiological                                               |
|                                                 | and biochemical parameters and survival rates                                                |
|                                                 | of the clam Anomalocardia flexuosa in response                                        |
|                                                 | to simulated MHWs. Under laboratory                                                          |
|                                                 | conditions, Anomalocardia flexuosa was allowed                                               |
|                                                 | to adapt to a stable control condition before                                                |
|                                                 | being exposed to simulated conditions of MHWs                                                |
|                                                 | occurrence lasting up to 21 days by warming the                                              |
|                                                 | tank water by 3°C above the control                                                          |
|                                                 | temperature."                                                                                |
| Comment4: L. 62-63: Could you elaborate on      | We have elaborated on this part (from line 84) as                                            |
| this further and explain the processes behind?  | follows: "Trophic dynamics of marine                                                         |
| this farther and explain the processes bening.  | ecosystems are affected by ocean temperature                                                 |
|                                                 | (Eddy et al., 2021; du Pontavice et al., 2019). In                                           |
|                                                 | particular, ocean warming is expected to                                                     |
|                                                 | increase the speed of energy transfer through                                                |
|                                                 | the food web, i.e., flow kinetic (du Pontavice et                                            |
|                                                 | al., 2020). Faster flow kinetic under ocean                                                  |
|                                                 | warming represents the increasing dominance                                                  |
|                                                 | of short-lived species so that each unit of                                                  |
|                                                 | biomass spends less time at a given TL and, on                                               |
|                                                 | average, across all TLs, ultimately leading to a                                             |
|                                                 | decrease in total biomass (Gascuel et al., 2008).                                            |
|                                                 | Simultaneously, ocean warming is expected to                                                 |
|                                                 | induce a decrease in biomass transfer efficiency,                                            |
|                                                 | altering both consumer production and biomass                                                |
|                                                 | due to larger energy losses between each TL (du                                              |
|                                                 | Pontavice et al., 2019). Therefore, ocean                                                    |
|                                                 | warming alters both the amount and speed of                                                  |
|                                                 | matter and energy transfer within the food web,                                              |
|                                                 | potentially leading to a decline in consumer                                                 |
|                                                 | biomass through independent and cumulative                                                   |
|                                                 | effects (du Pontavice et al., 2021; Guibourd de                                              |
|                                                 | Luzinais et al., 2023)."                                                                     |
| Comment5: L. 78: I would leave out "proceeding  | Yes, we agree, we have removed this part of the                                              |
| their occurrences", it makes the sentence       | sentence.                                                                                    |
| difficult to understand.                        |                                                                                              |
| Comment6: L. 89: Is there a specific reason for | Previous studies applying EcoTroph employed a                                                |
| using TL width = 0.1?                           | TL width of 0.1, primarily because of                                                        |
|                                                 | computational efficiency while maintaining a                                                 |
|                                                 | good representation of food web structure and                                                |
|                                                 | functions. We have mentioned this in the revised                                             |
|                                                 | manuscript (from line 119) as follows:                                                       |
|                                                 | "Practically, each biomass trophic spectrum with                                             |

|                                                                                                                                                                                                                                                                              | The above 1 is split into small treatis classes                                                                                                                                                                                                                                                                                                                                                                                                                                                                                                                                                                                                                                              |
|------------------------------------------------------------------------------------------------------------------------------------------------------------------------------------------------------------------------------------------------------------------------------|----------------------------------------------------------------------------------------------------------------------------------------------------------------------------------------------------------------------------------------------------------------------------------------------------------------------------------------------------------------------------------------------------------------------------------------------------------------------------------------------------------------------------------------------------------------------------------------------------------------------------------------------------------------------------------------------|
| Comment7: L. 92-93: Why is the biomass transfer in lower TLs faster?                                                                                                                                                                                                         | TL above 1 is split into small trophic classes bounded by pre-defined lower and upper trophic levels (with conventionally TL width = 0.1 - primarily because of computational efficiency while maintaining a good representation of food web structure and functions - in the steady-state version of EcoTroph."  It is due to the shorter life expectancy from low TL individuals. We have modified the sentence as follows (see lines 126-127): "The time needed for the biomass to flow from one to the next trophic class varies along the food chain, with biomass transfers generally faster in lower TLs (as species generally have short life-expectancy) than in the higher ones.". |
| Comment8: L. 97: biomass flows between(?) the trophic biomass spectra                                                                                                                                                                                                        | No, the correct term is "in" and not "between" as with EcoTroph-Dyn, we analyse biomass flow changes in each trophic biomass spectrum. By "each trophic biomass spectrum," we refer to the trophic biomass spectrum represented in each 1° × 1° longitude-latitude ocean cell.                                                                                                                                                                                                                                                                                                                                                                                                               |
| Comment9: L. 118: I don't understand what you mean by "the trophic level of each TL classes j.". Do you mean the trophic level of the j TL class?  Comment10: L. 119-120: Is this assumption based on observations/experiments? Give appropriate references.                 | Yes, we meant that, we have modified the sentence in the revised manuscript (see line 157).  We have modified the sentence as follows (line 158): "MHWs cause marine organism mortality, impacting their life expectancy (Smith et al., 2023). In EcoTroph-Dyn, these changes in life expectancy are reflected in the loss rate within the biomass spectrum, representing the proportion of biomass that neither persists nor progresses through the food web (du Pontavice et al., 2021; Gascuel et al., 2008, Guibourd de Luzinais et al., 2024). Therefore, according to equations used in the steady-state version of EcoTroph, the flow kinetics during MHWs is increased as follows:"  |
| Comment11: L. 125-126: "Biomass spectra in EcoTroph-Dyn are split into trophic classes with variable widths of trophic levels." – I don't really understand this sentence. Do you mean: "Biomass spectra in EcoTroph-Dyn are split into trophic classes of variable width."? | Yes, we meant that, we have modified the sentence in the revised manuscript (see line 167).                                                                                                                                                                                                                                                                                                                                                                                                                                                                                                                                                                                                  |
| Comment12: L. 164-165: "lasting 15 days of the fortnight" – something must be wrong here.                                                                                                                                                                             | Sorry it's a misuse of language. We ran the EcoTroph-Dyn model at a half month time step, and the expression seems to be a fortnight? We define a fortnight for our study as 1/24 of a year in line 219. To avoid confusion, we have changed                                                                                                                                                                                                                                                                                                                                                                                                                                                 |

|                                                                                                                                                                         | the term "fortnight" to "15 days" throughout the                                                                                                                                                                                                                                                                                                                                                                                                                                                                                                                                                                                                                                                                                                                                                                                                                                                                                                                                                                                                                                                                                                                                                                                                                                                                                                                                                                                                                                                                                                                                                                                                                                                                                                                                                                                                      |
|-------------------------------------------------------------------------------------------------------------------------------------------------------------------------|-------------------------------------------------------------------------------------------------------------------------------------------------------------------------------------------------------------------------------------------------------------------------------------------------------------------------------------------------------------------------------------------------------------------------------------------------------------------------------------------------------------------------------------------------------------------------------------------------------------------------------------------------------------------------------------------------------------------------------------------------------------------------------------------------------------------------------------------------------------------------------------------------------------------------------------------------------------------------------------------------------------------------------------------------------------------------------------------------------------------------------------------------------------------------------------------------------------------------------------------------------------------------------------------------------------------------------------------------------------------------------------------------------------------------------------------------------------------------------------------------------------------------------------------------------------------------------------------------------------------------------------------------------------------------------------------------------------------------------------------------------------------------------------------------------------------------------------------------------|
|                                                                                                                                                                         | manuscript.                                                                                                                                                                                                                                                                                                                                                                                                                                                                                                                                                                                                                                                                                                                                                                                                                                                                                                                                                                                                                                                                                                                                                                                                                                                                                                                                                                                                                                                                                                                                                                                                                                                                                                                                                                                                                                           |
| Comment13: L. 168: I don't understand this sentence. Do you mean: "We used an alpha of 0.2 in our simulations."?                                                 | Yes, we meant that, we have modified the sentence in the revised manuscript (line 213).                                                                                                                                                                                                                                                                                                                                                                                                                                                                                                                                                                                                                                                                                                                                                                                                                                                                                                                                                                                                                                                                                                                                                                                                                                                                                                                                                                                                                                                                                                                                                                                                                                                                                                                                                               |
| Comment14: Figure 1: How do you derive transfer efficiency, MHW mortality, and flow kinetic from satellite data?                                                 | To derive transfer efficiency, MHW mortality, and flow kinetic from satellite data, we used empirical equations where satellite data (SST) intervenes                                                                                                                                                                                                                                                                                                                                                                                                                                                                                                                                                                                                                                                                                                                                                                                                                                                                                                                                                                                                                                                                                                                                                                                                                                                                                                                                                                                                                                                                                                                                                                                                                                                                                                 |
| Comment15: L. 193-194: Why did you use a single threshold and not one for each month, for example? Which impact may the use of only one threshold have on your results? | The objective of the study was to assess the impact of MHWs, to do so we preferably detect the ones where species are undergoing a thermal stress which occurs mostly during summer period.  We have clarified the framework of the study by defining MHWs in the introduction as follows (see line 99): "This study aimed to disentangle the additional or synergistic consequences of MHWs occurring during the year's warmest month, when species are undergoing a thermal stress, with the effects of the slow-onset climate changes in marine ecosystems."  Furthermore, we have incorporated the following paragraph after line 735 in the discussion.  "Here, we focused solely on the direct impacts of MHWs occurring during the year's warmest month via thermal stress, resulting in species mortality (Oliver et al., 2021; Smith et al., 2023). However, MHWs in other seasons can also have consequences on populations by affecting a specific stage of the life cycle of certain species (Crickenberger & Wethey, 2018; Oliver et al., 2021; Smith & Thatje, 2013). For example, MHWs that stress adult breeders can lead to a decrease in reproductive investment and, consequently, fewer, smaller, and lower quality gametes (e.g., Shanks et al., 2020), resulting in a loss of abundance and biomass of some species (Johansen et al., 2021). While taking seasonality into account will increase the number of detected extreme events, some may not have ecological consequences (Oliver et al., 2021; Smith et al., 2023). Thus, our approach can be seen as conservative and may underestimate the impact of MHWs. Nevertheless, we detected MHWs with potentially significant ecological impacts. Studying MHWs occurring in seasons other than summer would involve considering the phenological effects of MHWs. However, |

since the EcoTroph-Dyn model does not directly represent this phenological aspect of marine organisms, future studies can apply other approaches that explicitly represent seasonal processes such as spawning and migration to elucidate the effects of phenology" In our study, we can have a MHW day with Yt < Comment16: L.215: How can a day with Yt < Tt + St be an MHW day? Can you explain this in Tt + St. This comes from the use of an annual more detail? threshold value to detect MHW events (see appendix figure S5 just before April month). We have rewritten this part for greater clarity as follow (see line 263): "In our study, we can have a MHW day declared even though Yt<Tt + St. This specific situation is rare (less than 0.5% of time series) and occurs because of the use of an annual threshold value to detect MHW events that mainly occurred during the year's warmest months. (See appendix figure S5 just before April month for schematic visualisation. The time series created using this algorithm is referred here as 'without MHW'." Comment17: L. 224-229: Could you further Yes, we have added these details. We developed explain the EPPLEY-VGM method? Not all the idea line 277 as follows: "The EPPLEY-VGPM readers may be familiar with this method nor the method is a hybrid model that employs the basic VGM method, so I think especially basic model structure and parameterisation of the information would be helpful. For example, what standard VGPM (Vertically Generalized is the general concept of these methods and Production Model) computation. This model what is Pb\_opt? estimates net primary production (NPP) based on chlorophyll concentration, incorporating the vertical distribution of primary production. The specificity of the EPPLEY-VGPM method is that the polynomial description of the maximum daily net primary production found within a given water column (Pb opt, expressed in units of mg carbon fixed per mg chlorophyll per hour) is replaced by the exponential relationship described by Morel (1991), based on the curvature of the temperature-dependent growth function described by Eppley (1972)." Comment18: L. 230-235: This should at least be We have added these sentences from line 767: "Secondly, in this study, in order to propose a mentioned in the Discussion (somewhere in the suitable representation of the world ocean, we paragraph I. 656-675). use an interpolation method to reconstruct an incomplete NPP time series. The interpolation was constrained by the minimum and maximum satellite data values of the NPP observed over their respective time series to ensure reliable interpolation and reduce potential bias."

| Comment19: L. 236-237: Which biases may be introduced by this duplication? This should also be included into Sect. 5.3.  Comment20: L. 285-287: "Furthermore, more                                                                                                                                       | We have added these sentences from line 770: "Thirdly and lastly, in this study, we duplicated NPP monthly value to able the EcoTroph-Dyn to run at a 15 days basis. This duplication may have smoothed marine ecosystem response to the historical changes in marine environment; However, it has not changed trends and conclusions of our results."  Yes, see response to comment4 of reviewer RC1.          |
|----------------------------------------------------------------------------------------------------------------------------------------------------------------------------------------------------------------------------------------------------------------------------------------------------------|-----------------------------------------------------------------------------------------------------------------------------------------------------------------------------------------------------------------------------------------------------------------------------------------------------------------------------------------------------------------------------------------------------------------|
| days with MHWs with lower intensity were identified for low latitude regions (23°N - 6°S) (Figure 3c) compared to MHWs identified in higher latitude regions (> 23°N and 25°S)." This sentence is not really clear. Do you mean that the intensity of MHWs was generally lower in high latitude regions? |                                                                                                                                                                                                                                                                                                                                                                                                                 |
| Comment21: Figure 3: The figure caption seems to be mixed up. The description for c) seems to match panel d), while the description for d) does not match any panel. Thus, panel c) has no fitting description.                                                                                   | Please, see response to comment4 of reviewer RC1.                                                                                                                                                                                                                                                                                                                                                               |
| Comment22: L. 304: "effects of the short-term impacts of MHWs on the long-term changes" This part seems a bit confusing and contradictory, I would leave out the "short-term impacts".                                                                                                                   | Yes, we agree, we have removed this part of the sentence (see line 372).                                                                                                                                                                                                                                                                                                                                        |
| Comment23: L. 320: Even if explained in the caption of Fig. 4, I would also define the three biomes in the text since the figure may be placed somewhere else in the typeset paper.                                                                                                               | Ok, we have added the biomes definition in the main text as follows (line 382):"While total consumer biomass was projected to decrease slightly across the three biomes (temperate, tropical, and upwelling biomes represented in Figures 4b, c, and d, respectively)".                                                                                                                                         |
| Comment24: L. 337: Using numbers for biomass increase for both scenarios would make it easier to compare the results, i.e., "a biomass decrease increase was projected to occur in 76%24% of the global ocean"                                                                                           | We agree, we have modified the sentence in the revised manuscript, see line 420.                                                                                                                                                                                                                                                                                                                                |
| Comment25: L. 358: Maybe it would be useful to give the number for global biomass loss without MHWs again for direct comparison.                                                                                                                                                                         | We agree, we have added the percentage in the revised manuscript and modified the sentence as follows (line 422): "Under the 'with MHWs' scenario, this proportion decreased and a biomass decrease was projected to occur in 76% of the global ocean, with a projected global biomass decrease of 4.8% by 2015-2021 relative to 1998-2009 compared to an only 2.4% biomass loss under 'without MHW' scenario." |
| Comment26: L. 365: What do you mean with "expected"? You already analyzed the differential impact of MHWs on trophic levels in Sect. 4.2.2, didn't you?                                                                                                                                                  | Yes, we have changed the wording "were expected" by "are projected", see line 429.                                                                                                                                                                                                                                                                                                                              |

|                                                                                                                                                                                                                                                                                                                                                           | T                                                                                                                                                                                                                                                                                                                                                                        |
|-----------------------------------------------------------------------------------------------------------------------------------------------------------------------------------------------------------------------------------------------------------------------------------------------------------------------------------------------------------|--------------------------------------------------------------------------------------------------------------------------------------------------------------------------------------------------------------------------------------------------------------------------------------------------------------------------------------------------------------------------|
| Comment27: L. 377: Can you explain what an ANOVA is?                                                                                                                                                                                                                                                                                                      | Sure, we proposed to add an explanation in the Material & Methods section line 218: "We performed a three-way ANOVA, that is a statistical test we used to analyse the effects of trophic levels, biomes, and $\alpha$ value on biomass change."                                                                                                                         |
| Comment28: L. 383: greatest instead of greater?                                                                                                                                                                                                                                                                                                    | No, the correct word is greater.                                                                                                                                                                                                                                                                                                                                         |
| Comment29: L. 403: Why do you use different reference periods to define temperature anomalies? In this way, the anomalies are not consistent. I would suggest to choose one of both periods. Did you use the same reference periods to calculate the temperature anomalies shown in Fig. 8? If yes, these should be corrected as well.                    | We always used the reference period 1982-2011 to define temperature anomalies. We have rephrased the sentence (line 467) as follows "Temperature anomalies were on average >=4 °C (between 2013 to 2016) and up to 8 °C in 2015 relative to 1982-2011." to enhance clarity                                                                                               |
| Comment30: L. 430-431: This part is difficult to understand. Do you mean: "Considering the influence of MHWs from 2013 to 2016 using the 'with MHWs' scenario and alpha=0.2 resistance capacity"?                                                                                                                                                  | We agree, we have modified the sentence in the revised manuscript, see line 495.                                                                                                                                                                                                                                                                                         |
| Comment31: L. 456-459: This sentence is difficult to understand. Maybe replace with something like: "In this study, we accounted for MHWs in the last four decades using hindcast simulations and showed the potential of synergic impacts of MHWs (pulses) and long-term climate change (presses) on bio-mass and trophodynamics of ecosystems.". | We agree, we have modified the sentence in the revised manuscript, see line 521.                                                                                                                                                                                                                                                                                         |
| Comment32: I. 494-495: This explanation of TE would be helpful in the methods section.                                                                                                                                                                                                                                                             | We agree, and have moved it to the method section, see lines 145-150.                                                                                                                                                                                                                                                                                                    |
| Comment33: L. 521: metabolic efficiency or transfer efficiency? Shouldn't the ratio between ingested and stored energy be high, i.e., only a small part of the ingested energy is stored and the rest is lost?                                                                                                                                     | Thank you for pointing out this mistake, we have modified the sentence line 591 as follows: "Tropical ecosystems are composed of species with low transfer efficiency (TE) (low ratio between stored energy and ingested energy), with significant energy losses that increase with temperature (Brown et al., 2004; du Pontavice et al., 2020; Schramski et al., 2015). |
| Comment34: L.529-530: Why is the mortality higher in low TLs?                                                                                                                                                                                                                                                                                      | We have added on line 600 "as low TLs tend to have lower thermal limit than high TLs" to enhance clarity.                                                                                                                                                                                                                                                                |
| Comment35: L. 534: If the increase is 1% I wouldn't use the word "sharp".                                                                                                                                                                                                                                                                          | We agree and have removed the word sharp in the revised manuscript (see line 604).                                                                                                                                                                                                                                                                                       |
| Comment36: L. 561-562: I don't understand part (ii), could you explain this further?                                                                                                                                                                                                                                                               | We have modified this part as follows (line 632-637): "(ii) the specific functioning of these ecosystems with cool water rising from depth to the surface, tends to reduce the number of MHW days compared to their adjacent open ocean (Varela et al., 2021). More generally, it                                                                                        |

|                                                                                                                                                                                                                                                                        | has been highlighted that ocean warming does
not affect coastal regions with upwelling in the
same way as the open ocean (Varela et al.,
2021)."                                                                                                                                                                                |
|------------------------------------------------------------------------------------------------------------------------------------------------------------------------------------------------------------------------------------------------------------------------|------------------------------------------------------------------------------------------------------------------------------------------------------------------------------------------------------------------------------------------------------------------------------------------------------------------------------------------|
| Comment37: L. 585-587: The structure of this sentence seems odd and makes it difficult to understand. Please check and revise.                                                                                                                                  | Thank you, we have revised the sentence as follows (see line 660): "In contrast, in the California Current biogeochemical region, the MHW was associated with a substantial increase in the abundance of pyrosomes which implied a limitation of energy flow moving toward higher trophic levels has been observed (Gomes et al., 2024)" |
| Comment38: L. 599-600: What defines the models in this family? What do they have in common?                                                                                                                                                                            | We have rephrased the sentence (see line 676) to enhance clarity as "Despite its apparent simplicity and the reduced number of parameters, EcoTroph-Dyn is part of the family of "complete ecosystem models and dynamic system models" (Plagányi, 2007) as it represents all trophic levels, from primary producers to top predators."   |
| Comment39: L. 607-608: What are biomass                                                                                                                                                                                                                                | Mistake in the wording, it is simply biomass: we                                                                                                                                                                                                                                                                                         |
| density values?                                                                                                                                                                                                                                                        | have deleted density (see line 683).                                                                                                                                                                                                                                                                                                     |
| Comment40: L. 615: The word "projection" usually refers to simulations/estimates for the future. Since you performed hindcast simulations, I would use a different word here to avoid confusion. Please also check the rest of the manuscript.                         | We have changed it to "historical simulation", see line 693.                                                                                                                                                                                                                                                                             |
| Comment41: L. 629-632: Can you quantitatively compare your results to those of Arimitsu et al. (2021)?                                                                                                                                                                 | We can only compare qualitatively our results with those of Arimitsu et al., 2021, as in their study they work at the species level and did not account for species of the entire foodweb/model, while in our study we used a trophic level-based approach.                                                                              |
| Comment42: L. 640: The reference Cheung et al. (2020) does not exist in your reference list. Do you mean Cheung & Frölicher (2020)?                                                                                                                                    | Yes, thank you for pointing out this mistake.                                                                                                                                                                                                                                                                                            |
| Comment43: L. 647-448: This part is difficult to understand. Maybe replace with "It would therefore have been valuable to test EcoTroph-Dyn against other MHWs in the world ocean".                                                                                    | Yes, your proposition is simpler, we havel changed it, see line 725.                                                                                                                                                                                                                                                                     |
| Comment44: L. 681: What do you mean by "dismiss any possibility"?                                                                                                                                                                                                      | We have rewritten the sentence (line 781) as follows: "To be cautious, we considered various loss rate scenarios to obtain a complete range of responses from marine ecosystems."                                                                                                                                                        |
| Comment45: L. 684-688: This sentence is quite complex and difficult to understand. Maybe replace with something like: "Even though the global impact of MHWs is negative, species-explicit modelling could improve our understanding of how various impacts of climate | We agree and have made changes to the revised version of the manuscript (see lines 787-790).                                                                                                                                                                                                                                             |

change and species-level responses will affect trophodynamics and ecosystem structure and function.". Comment46: Sect. 5.4: You could highlight here Yes EcoTroph-Dyn model can be used to project how future work can build on your study in the impacts of hypothetical future MHWs at particular. For example, what analyses should global and local scales. Please, see the response your model be used for in the future? Should to reviewer RC1 comment5. your model be modified/extended, and if so, how? **Technical corrections** Throughout the manuscript, there are some Thank you for pointing out these technical issues with reference formatting (i.e., the use of corrections line by line. We have addressed them through the revised version of the parentheses, commas, and semicolons). I have already included a few examples below. manuscript and make a full check of the manuscript for others. L. 11: are becoming have become L. 20: (NPP) data L. 21: observations L. 22: by trophic levels L. 25: specific MHW-induced decline in biomass of  $8.7\% \pm 1.0$  (standard error) in the region from 2013 to 2016. L. 27: than in lower L. 36: resulting in more than a doubling of the number of MHWs days L. 37: a space is missing before the reference L. 43: have caused a decrease L. 50: simulationnumerical modelling L. 51: Don't use a comma for in-text citations: Carneiro et al., (2020) L. 66: function globally have not yet been

clearly understood on a global scale

- L. 68: climate changes
- L. 69 and 1. 70: Since this is an article with multiple authors, I would use "we" consistently.
- L. 70: MHWs (see (Guibourd de Luzinais et al., 2024)
- L. 77: occurred in on the
- L. 79: Material and methods
- L. 83: from by primary producers
- L. 85: food webs
- L. 87: TLs, i.e.,
- L. 88: trophic spectra spectrum
- L. 90: the whole consumers biomass
- L. 92: generally being faster
- L. 94-95: the references should be put into parentheses
- L. 96: ecosystems biomass
- L. 102: MHWs occurrence. EcoTroph-Dyn's algorithms' details
- L 112: TL. year-1
- L. 114: trophic level, using
- L. 131: within the TL class [τ,τ+Δτ[. It], is expressed as
- L. 144: dependent dependent
- L. 147: flow kinetic (K)) and where
- L. 149: 3.2 MHW loss rate algorithm computation

- L. 152: loss rate algorithm computation
- L. 153: into EcoTroph-Dyn
- L. 154: period (period 1982-2011)
- L. 156: Matching historical MHWs'
   historical distributions and
   characteristics with species distribution
- L. 158: Estimation bBased on this percentage estimation of an additional loss rate
- L. 159-160: Finally, through loss rate (ηi)
  mathematical expression, we assumed in
  the mathe-matical expression of loss rate
  ηi that species arewere continuously
  challenged by MHW in-creased MHW
  intensity, which is expressed as:
- L. 164-165: with β rangesranging from β=0; (no MHW), to β=1; (MHW lasting 15 days of the fortnight)
- L. 165: MHWcat,i corresponds to an MHW intensity index
- L. 169: to community resistance capacity
   to MHW by testing
- L. 184: without MHWs
- L. 189: every MHWs day
- L. 207: seasonal component (*St*), is then
- L. 208: estimation of (*St*) on the trendadjusted series

- L. 212-213: without MHWs
- L. 215: component, i.e.,
- L. 215: For MHW days with Yt below (Tt + St) or not an MHW daynon-MHW-days, we keep
- L. 217: referred to here as
- L. 218: match adapt
- L. 222: when an MHW lasts for an entire fortnight.
- L. 225: is a hybrid model
- L. 239-240: under for the scenarios
- L. 246: of simulating 12 years
- L. 254: past MHWs events
- L. 267: by in the period 2015-2021
- L. 268: NPP changes were was observed
- L. 268: Notably In particular
- L. 270, 272: in the period 2015-2021
- L. 273: warmed up warming by 1°C during over
- L. 277: relative to the average between and the average of
- L. 292: Evolution of the spatial extent
- L. 292-293: Evolution of MHWs averaged duration categorised by their intensity
- L. 299: on average, by  $0.07 \pm 0.02\%$

- L. 317: S2 for biomes spatial definition
- L. 321: with MHWs', the declines
- L. 332, L. 339, L. 358, L. 375: by in the
   period 2015-2021
- L. 335 and Fig. 5 caption: For the trophic level classes, the second opening parenthesis needs to be a losing parenthesis
- L. 336-337: notably with the tropical and upwelling biomes being notably more impacted.
- Fig. 5: Change in trophic groups biomass (y-axis)
- L. 345-346: Projected changes in consumer biomass by trophic levels and biomes under the 'without MHW' and 'with MHWs' scenarios relative to the 1998-2009 average between 1998-2009.
- L. 362: off the coast of Papua New Guinea Coast
- L. 389: by trophic levels
- L. 392: 75th quantiles
- L. 395: the response of low TLs response
- L. 402: in the biomass spectrum
- L. 403: relative to the 2016 average
- L. 416: exhibited a significant total consumer biomass decrease

- L. 417: the scenarios with and without
   MHWs
- L. 420: the most
- L. 420-421: However, uUnder
- L. 425: provinces were the most affected
   by the MHW
- L. 426: biomass decreases of 5% and
   3.8% 'with MHWs' relative to the
   'without MHW' scenario
- L. 429: lower TLs
- L. 433: as of by 2021
- L. 443: change in the 'without MHW' and 'with MHWs' scenarios
- L. 444: indicates the duration of 'the Blob' duration.
- L. 451: 0.2, while
- L. 452: by in the period 2013-2016
- L. 457-458: longterm long-term
- L. 466, 468: Be careful with the use of past tense. The suggestions of your study have not expired, so use "suggest" instead of "suggested" in L. 466. Similar cases appear throughout the manuscript
- L. 470: ecosystems, which is congruent with studies by Arimitsu et al., (2021);,
  Gomes et al., (2024);, and Smith et al.,
  (2023) studies.

- L. 373: ecosystems perturbations
- L. 475: of the perturbation in ecosystem functioning perturbation
- L. 476-477: intensity and duration of MHWs characteristics have continuously increased
- L. 478: hindcast period hindcast
- L. 482: recover to pre-perturbed
- L. 484: upwelling biomes, where the hindcast biomass of high TLs consistently
- L. 488: may, therefore, be
- L. 490: continuing continued
- L. 514: , which is used
- L. 551: These MHWs
- L. 564: through our modelling approach in our simulations
- L. 566: in the California Ocean
- L. 568: with anthe increase
- L. 570: differently differentially
- L. 575: Differences in exposure to the intensity
- L. 577: subjected
- L. 592-595: For example, communities in the Gulf of Alaskan Gulf are more efficient than those in the Californian Current (du Pontavice et al., 2020), and

the energy entering the food web was less disrupted than in the Californian Current, which may explain the greater impact of the MHW on the Californian Current.

- L. 598: MHWs hindcast
- L. 602: by their trophic levels
- L. 632-634: It is worth noting that projections obtained fromusing a smaller (larger) α led to an underestimation (overestimation) of biomass losses and changes in biomass flow parameters relative to the estimates of Cheung & Frölicher, (2020) and Gomes et al., (2024) esti-mates.
- L. 640-641: Grey violin plots correspond to results from Cheung et al., (2020), while the red ones corresponds to our hindcasted EcoTroph-Dyn simulation with α=0.2 scenario.
- L. 656: Furthermore, uncertainties
   aboutin our results arise from EcoTroph Dyn the environmental drivers of EcoTroph-Dyn.
- L. 657: EcoTroph-Dyn has been was driven

- L. 660: In this study, we did not consider the 'with' and 'without' MHWs scenarios for NPP.
- L. 663: couldmay have overestimated

  MHWs impacts
- L. 666: Why do you capitalize marine ecosystem models?
- L. 681-682: Running the aforementionedthese five MHW-induced loss rate-induced scenarios
- L. 683-684: with a worsening an increasing biomass loss over marine ecosystems with de-creasing resistance capacities decreasing (increasing α increasing).
- L. 690: From In our study
- L. 693: anomalously low wind, an anomalously weak Ekman transports
- L. 694: north, and, coupled with anomalously low air-sea heat exchanges
- L. 696: have already contributed
- L. 700: better understand better

---

## Referee Report (RR1)

This is a novel study that uses modelling to estimate the impacts of MHWs on ecosystem functioning at a global scale. The methodology is explained clearly, and the results are presented and discussed effectively. There are still minor grammatical errors throughout the paper that affect clarity and readability. I have listed my suggestions line by line below.

Line 35 - "periods of more than 5 days of"

Line 59 - "drive" or "result in" would be more accurate than "imply"

Line 60 - would be clearer with a comma between "invertebrates" and "modifying"

Line 73 - "employed" is repeated twice

Line 75 - "implications" should be plural

Line 100 - "undergoing thermal stress"

Line 117 - "web's functioning"

Line 138, line 197 - "we provide"

Line 165 - "are increased" or "increase"

Line 170 - the second bracket should be a closed bracket

Line 258 - lower case "the"

Line 274 - "15-day time step"

Line 295 - "two sets of 15 days" or similar

Line 360 - "average duration of MHWs"

Line 394 - "1.0 ± 0.1%" - you should consistently use the same number of decimal places

Line 445, 447, 483, 786 - "-16" should be superscript

Line 468 - "from 2013 to 2016"

Line 477, 649, 786 - "p-value"

Line 522 - "synergistic" is more widely used in ecological literature; see doi:

10.4319/lo.1999.44.3\_part\_2.0864

Line 535 - "analysis suggests" or "analyses suggest"

Line 540 - "and Smith et al. 2023"

Line 601 - "thermal limits"

Line 605 - "an acceleration"

Line 623 - "moderate mortality in these ecosystems" (to clarify that you are referring specifically to upwelling systems, not the effects of MHWs at a global scale)

Line 634 - "which tends"

Line 694 - "biomass change simulations"

Line 702 - "MHW impacts"

Line 701 - "specific regions"

Line 704 - "historical biomass simulations" would sound better

Line 712 - "availability and quality"

Line 715 - the last word, "estimates", should be removed from the sentence

Line 726 - quotation marks are not needed

Line 758 - "MHW impacts"

Line 772 - "monthly values to enable the EcoTroph-Dyn to run with 15-day timesteps"

Line 773 - "ecosystem responses"

Line 774 - "however" should be lowercase

Line 781 - "MHW occurrence"

---

## Referee Report (RR2)

**Review after Major Edits: Marine heatwaves deeply alter marine food web structure and function**

**Summary:**

This paper analyzes the influence of marine heat waves (MHWs) on ecosystem dynamics using the EcoTroph-Dyn model. This dynamic model is driven by satellite-based estimates of NPP and SST, and by removing MHWs from the SST dataset, the authors compare and contrast the results of a the model (eg. biomass flow, transfer efficiency) with and without the influence of MHWs. Adding clarifications suggested by previous reviewers has improved this paper from the previous version.

They find, and clearly present, that that highest declines in biomass occur in higher trophic levels, and the tropical biome was most impacted (largest decrease in biomass and highest sensitivity). They adequately discuss the assumptions and uncertainties in their analyses, and noted some possible next steps.

**General/specific comments:**

Generally reads well. Some sentences became confusing when using parameters that had not yet been defined, or using symbols/names inconsistently. I also had trouble with some plural/singular versions of words (kinetic flow vs kinetic flows), which might be eased if parameters symbols/shorthands (e.g. K) were used consistently (?). Below are some minor technical edits by line, with a few minor questions for clarification.

How does EcoTrophDyn determine how many Trophic Levels are included in each biome?

- 18 is the years warmest month the same each year?
- 35- defined as a period of more than 5 days of....
- 47 related to impacts on their body temperature
- 48 within a range of body temperatures
- 49 certain
- 50 exceed the temperature optima
- 59 not sure what you mean by "imply" here?
- 74 employed (written twice)
- 80 years and location of "the blob" or reference "the blob" in the intro paragraph 1.

```
81- MHW, no plural
84 – the ecosystem modelling framework
87,88 – haven't explained flow kinetic, and it feels like it needs a plural!
89 – trophic level (TL), I don't think you've introduced TL abbreviation yet
100 - a
100 - change "with" to "from"
116 – no comma
116 – add comma (trophic level, TL = 1)
122 – with conventional TL width = 0.1...
129 – remove a comma, move the parenthesis
       ....fishing (e.g., du Pontivice et al., ....)
133 - ...a 15-day time step
133 - change "as" to "because"
152 - ...flow kinetic parameter, Kt.... Or at least put a comma
152 – I'm confused about whether you actually have flows (plural) to assess in each spectrum or
you really mean flow (also line 132) like the flow kinetic.
165 – are
178 - is this the right bracket?
183- one, I think convention is write out every number below 10? Not sure here
Figure 1 -
214 – didn't define this alpha term for us yet.
227
238
314
384 – hard to see the decline in topical biome> than world because the graphs are on different
```

scales....

Figure 5 – I'd almost rather see this figure plotting single TL groups for both with and without MHW, then you'd be able to see the change caused by the MHW. In this version it is harder to scan across two panels to see how different the red lines are....

Why are tropical biomes more sensitive?

467 – Figure 8a is of temperature not biomass spectra

472 – wrong figure panel again

479 - wrong figure panel #

535 - plural

554 – so why was the tropical biome TL biomass more permanently affected? Do tropical foodwebs have a slower turnover rate?? Are warm water organisms already living closer to their metabolic limits? (sort of addressed in discussion 591+)

579 - do you mean disrupt catabolic processes and be undetectable (and/or counfounding) in the anabolic-based TE measure?

579 - flow kinetic is a singular parameter here? (again line 584)

580 – transfer

588 – I don't see an S10

595 – is rapid biomass flow helpful for ecosystem recovery after MHW impacts?

640 – I think two commas would help readability

...species, and the growth of populations, adapted..."

694 - scenario?

726 - remove "

814 - response to MHWs of species?